# Rational strain design with minimal phenotype perturbation

Bharath Narayanan[1,3], Daniel Weilandt[1,4], Maria Masid [2], Ljubisa Miskovic [1] ✉ & Vassily Hatzimanikatis [1] ✉

Devising genetic interventions for desired cellular phenotypes remains challenging regarding time and resources. Kinetic models can accelerate this task by simulating metabolic responses to genetic perturbations. However, exhaustive design evaluations with kinetic models are computationally impractical, especially when targeting multiple enzymes. Here, we introduce a framework for efficiently scouting the design space while respecting cellular physiological requirements. The framework employs mixed-integer linear programming and nonlinear simulations with large-scale nonlinear kinetic models to devise genetic interventions while accounting for the network effects of these perturbations. Importantly, it ensures the engineered strain's robustness by maintaining its phenotype close to that of the reference strain. The framework, applied to improve the anthranilate production in *E. coli*, devises designs for experimental implementation, including eight previously experimentally validated targets. We expect this framework to play a crucial role in future design-build-test-learn cycles, significantly expediting the strain design compared to exhaustive design enumeration.

Advances in gene editing techniques and the ever-increasing availability of omics data have spawned intense efforts in metabolism research. Within the biomedical domain, this has enabled us to glean broader insights into the metabolic phenotypes of various diseases, allowing for more informed therapeutic interventions[1–3]. In biotechnology, these advances have led to the creation of environmentally friendly, cost-effective bio-foundries using genetically engineered cellular organisms for optimal production of valuable compounds[4]. These metabolic engineering undertakings are typically implemented as a design-build-test-learn cycle, involving multiple experimentation stages and fine-tuning strain designs[5].

While technological advances have facilitated the genetic manipulation of organisms, significant challenges remain in determining the targets and the extent of such manipulations. Since robustness to changing environmental conditions is essential for the viability of designed strains, we need to ensure that the genetic

interventions maintain critical cell properties such as the energy charge and redox potentials[6–9]. Developing strategies targeting more than one enzyme is typically necessary to achieve this. Unfortunately, devising such multi-target strategies by direct experimentation requires considerable time and resources. One approach to reducing these costs is to conduct rational metabolic engineering using computational models to narrow down the range of strategies to be experimentally verified.

In particular, dynamic metabolic models are well suited for this task since they can capture the temporal evolution of the metabolic states to environmental and genetic perturbations under real-world fermentation conditions. However, the lack of available information about the values of kinetic parameters hampers the development of these models. Indeed, even for well-studied organisms such as *E. coli* or *S. cerevisiae*, we can find experimentally obtained values for only a few parameters in the literature and databases[10,11]. To infer the values of

[1]Laboratory of Computational Systems Biology (LCSB), Ecole Polytechnique Fédérale de Lausanne (EPFL), CH-1015 Lausanne, Switzerland. [2]Ludwig Institute for Cancer Research, Department of Oncology, University of Lausanne, and Lausanne University Hospital (CHUV), Lausanne, Switzerland. [3]Present address: Department of Oncology, University of Cambridge, Cambridge CB2 0XZ, UK. [4]Present address: Lewis-Sigler Institute for Integrative Genomics, Princeton University, Princeton, NJ 08544, USA. ✉e-mail: ljubisa.miskovic@epfl.ch; vassily.hatzimanikatis@epfl.ch

missing kinetic parameters, researchers have traditionally employed parameter estimation[12–14] and Monte Carlo techniques[15–19]. Recently, there have been numerous efforts to use machine learning to accelerate the building of these models[20–23].

Even when a high-quality kinetic model is available, it is computationally challenging to determine targets for metabolic engineering that meet desired design specifications because it requires simulating the metabolic network's responses for many putative designs. For example, to explore all possible strategies for manipulating (increasing or decreasing) the activities of five enzymes within a middle-sized metabolic network of 200 reactions (catalyzed by 200 enzymes), exhaustive enumeration would require performing more than $8.3 \cdot 10^{10}$ simulations. Additionally, for all these simulations, we would need to analyze whether or not the designed strains meet the specifications and preserve the robustness of wild-type strains exposed to long-term evolutionary pressure[24]. Hence, to perform reliable and comprehensive strain designs, the research community needs systematic, resource-efficient approaches that leverage the predictive capabilities of nonlinear kinetic models.

In this work, we report NOMAD (NOnlinear dynamic Model Assisted rational metabolic engineering Design). This computational framework scouts the space of candidate metabolic engineering strategies for those that satisfy the desired design specifications while preserving the robustness of the original phenotype shaped through evolutionary pressure and selection. As has been hypothesized and shown earlier[25–27], we achieve this by maintaining their metabolite concentrations and fluxes close to those of the reference strain. The rationale of trying to ensure a minimal deviation of the engineered strain phenotype from that of the reference strain has also been put forth in a constraint-based modeling approach called MOMA[24]. In this work, we go beyond MOMA, which only constrains metabolic fluxes to stay close to a reference strain. Our approach, based on kinetic models, allows us to impose constraints not only on fluxes but also on metabolite concentrations and the changes in enzyme levels. This results in a more accurate representation and design of the studied organisms, enabling us to capture both their steady-state and dynamic metabolic behaviors with greater fidelity. Additionally, NOMAD proposes testing the sensitivity and performance of the designs in nonlinear dynamic bioreactor simulations that mimic real-world experimental conditions. We can then rank the designs and suggest the best performers from these tests for experimental validation with high confidence. We validate NOMAD through two studies aimed at improving anthranilate production in previously studied *E. coli* strains[28]. We identify two sets of designs, each comprising five candidate designs, that are robust across phenotypic and expression uncertainty while providing superior in-silico performance compared with experimentally devised strategies[28]. Through its conception, NOMAD lends itself well to the DBTL (design – build – test – learn) cycle, with every round of iteration improving the quality of the proposed strain designs. Overall, this framework has the potential to accelerate the pace at which strain design breakthroughs are achieved, representing a potent disruptor within the biomedical and biotechnological domains.

## Results
### NOMAD for reliable strain designs
The NOMAD workflow consists of three steps (Fig. 1). It starts by generating a population of putative kinetic models consistent with experimentally observed omics and cultivation data, physicochemical laws, network topology, and regulatory interactions. These kinetic models consist of a system of nonlinear ordinary differential equations (ODEs) characterized by a set of kinetic parameters. To generate such models, we can use traditional kinetic modeling approaches such as ORACLE[15,18,29], K-FIT[30], Ensemble Modeling[19], MASSPy[17], and machine-learning empowered methods such as iSCHRUNK[20,22], REKINDLE[21], and RENAISSANCE[31].

In the second step of NOMAD, we perform several quality checks on the kinetic models and identify those that will ensure reliable in-silico strain design strategies[32]. In this screening process, we retain kinetic models that are (i) consistent with experimentally observed steady-state values of metabolic fluxes and metabolite concentrations, (ii) locally stable around that steady state; (iii) able to reproduce the dynamic behavior of metabolic responses under industrial production conditions; (iv) consistent with any available information on studied phenotype or a piece of expert knowledge; and (v) robust, meaning that these models resist change and are capable of coping with various genetic and environmental perturbations. This phase can be implemented using simulation and analysis tools such as SKiMPy[33], COPASI[34], and libRoadRunner[35].

In the final step, we use the screened models to design engineering strategies for achieving a chosen metabolic objective, such as the overproduction of high-value biochemicals. We use Network Response Analysis[26] (NRA) to perform the strain design. NRA casts the strain design process as an optimization problem that uses the outputs of the kinetic models ("Methods" section) and integrates design constraints ranging from the allowable fold changes in concentrations and fluxes to the extent and number of enzymatic interventions. This way, we obtain a computationally efficient modus operandi to enumerate designs and maintain the physiology of the engineered strain close to the reference physiology through various constraints.

After enumerating alternate strategies, we test their performance and sensitivity to phenotype and expression variability in fermentation simulations that emulate real-life conditions. Based on these tests, we rank the designs and propose the best-performing ones for experimental implementation. Additional NOMAD details are available in the "Methods" section.

As a case study for testing and validating the NOMAD workflow, we designed strategies for increasing anthranilate production in *E. coli* strain W3110 *trpD9223*. In an earlier experimental study, the strain was used as a scaffold for overproducing anthranilate through several genetic manipulations[28]. Subsequent subsections cover the results of the validation studies.

### Kinetic models calibrated to *E. coli W3110 trpD9923* physiology
We used ORACLE[15,16,36,37], implemented in the SKiMPy toolbox[33], to generate a population of 800,000 putative kinetic models that satisfied the experimentally observed steady-state behavior of the reference strain ("Methods" section). However, not all of these models were necessarily suitable for strain design due to poor dynamic characteristics or poor responses to engineering interventions, necessitating the process of model screening. We first screened the population of kinetic models for those with dominant time constants, quantified by the inverse eigenvalues at the steady state, at least 5× faster than the doubling time of the cell ("Methods" section). This meant that all metabolic processes were *expected* to settle into their steady states within the doubling time of the cell ("Methods" section). More than 11% of the generated models (91,852) showed such dynamic characteristics (Fig. 2a).

Whereas the inverse eigenvalues are a good indicator of the dynamic of metabolic responses in close vicinity of the steady state, due to the nonlinear nature of the system, not all models will exhibit the same dynamic behavior in a batch fermentation setting where metabolic states change intensely. Therefore, we tested if these models could reproduce experimentally observed behavior in a batch reactor. Out of 91,852 models, 212 captured experimentally observed temporal evolutions of growth, anthranilate, and glucose (Fig. 2b–d).

Next, we evaluated the suitability of these 212 models for strain design by testing their responses to naturally occurring random perturbations in enzyme activities. 10 out of the 212 models proved to be robust and consistent with the studied strain, exhibiting at least 50% of the growth of the reference strain ("Methods" section) when subjected

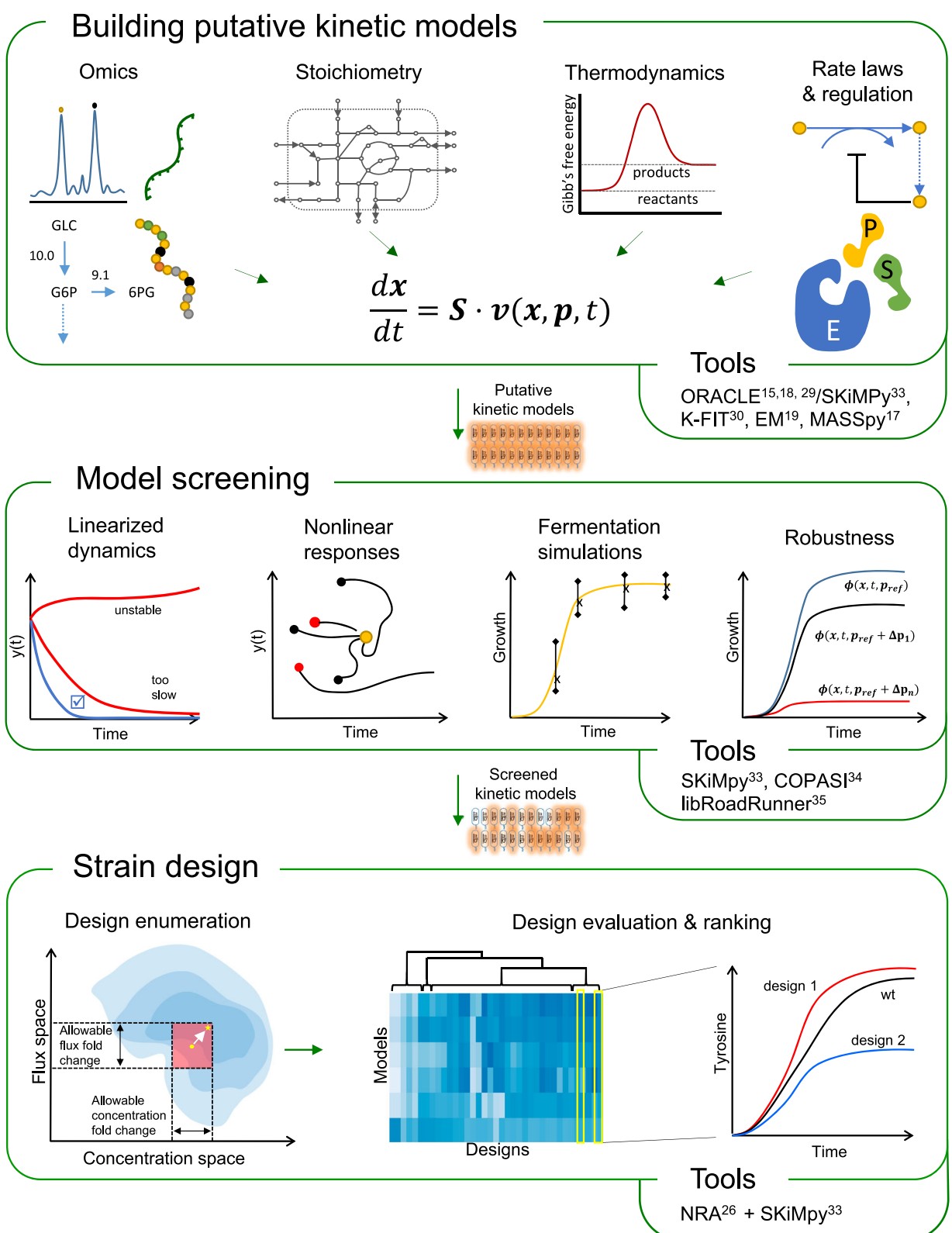

**Fig. 1 | NOMAD overview and required tools for each step.** We first integrate different types of data to build a set of putative kinetic models, represented by a system of ODEs. Next, we choose models based on dynamic characteristics such as their stability, ability to reproduce experimental fermentation data, and robustness to enzymatic interventions. Finally, we use the chosen models to conduct strain design. This involves solving a MILP optimization problem and enumerating designs that maintain the engineered strains close to the reference strain, evaluating the performance of these designs, and ranking them for experimental implementation.

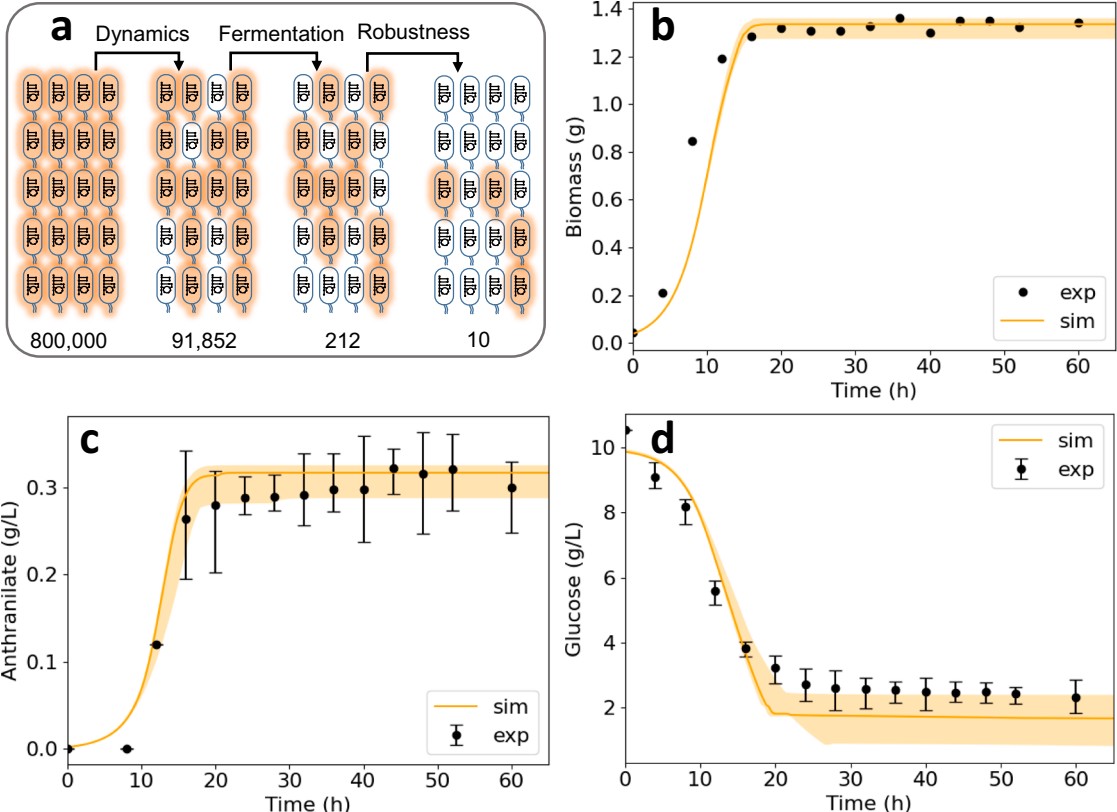

**Fig. 2 | Results of the model screening process. a** We filtered the 800,000 putative kinetic models down to 10 models that possessed the desired linearized dynamic characteristics, reproduced experimentally observed fermentation curves, and proved robust to enzymatic perturbations. **b–d** Comparison of the simulated growth (**b**), anthranilate (**c**), and glucose (**d**) responses of the 10 models with experimental batch fermentation data. The orange solid line and the shaded region represent the median and the interquartile range of the simulated responses respectively. The solid circles and error bars are the mean and standard deviation from the triplicate experiments. The behavior of the 10 kinetic models under batch fermentation conditions showed close agreement with the experimental data. Source data are provided as a Source Data file.

to these perturbations. These 10 models will be collectively referred to as K_trpD9923. The responses of K_trpD9923 in a batch fermentation setup show qualitative and quantitative agreement with the experimentally observed time evolutions of growth, anthranilate production, and glucose consumption (Fig. 2).

**Closeness to reference physiology for more reliable designs**
Extensive metabolic engineering might steer the engineered strain towards metabolic states with impeded growth or performance as, too often, the objective of overproduction of a metabolite has a major tradeoff with the organism's growth and global biosynthetic processes. For example, in efforts to optimize the specific productivity or yield of target chemicals, metabolic engineering interventions could inadvertently reduce other cell capabilities, such as ATP production or redox potential, by redirecting carbon and biosynthetic resources toward the target pathways and products. Because reference strains have evolved to maintain healthy and robust physiology, we postulate that we can engineer a productive strain by redirecting the flux to the desired objective while remaining as close to the reference state as possible. Thus, we use the proximity of the metabolic and fluxomic profile of the engineered strain as a proxy for maintaining a vigorous phenotype[25,26,38]. A related concept was also studied in the context of steady-state flux analysis[24]. Here, NOMAD uses nonlinear kinetic models and network response analysis (NRA, see "Methods" section) to implement this concept and constrain the phenotype perturbation while maximizing productivity.

We examined the impact of the phenotype perturbation constraint on the design performance by designing several sets of strains with an improved yield of anthranilate on glucose for each model in K_trpD9923. The sets differed by how much the engineered strains could deviate from the reference strain, quantified through a fold-deviation of the intracellular metabolite concentrations and metabolic fluxes from their values at the reference state. The set of strains closest to the reference strain could have intracellular concentrations deviating 2-fold from the reference strain. In contrast, the less constrained set could have intracellular concentrations deviating 20-fold from the reference strain. For all the sets, we allowed up to three enzyme modifications with a maximum of 5-fold change in their activities.

Current approaches to strain design using kinetic modeling perform a metabolic control analysis (MCA)[39,40] around the reference state and rank the target enzymes using the absolute value of the product flux or yield control coefficient with respect to each enzyme in the network. This approach does not consider constraints that could maintain the healthy physiology we discussed above. To understand the implications of using unconstrained MCA for strain design, we also applied a 5-fold change in enzyme activities to the enzymes corresponding to the top 3 anthranilate yield control coefficients for each kinetic model, without any constraints on concentration and flux perturbations.

The nonlinear responses of all these engineered strains showed that the closeness to the reference strain impacted performance (Fig. 3, Supplementary Note 1, Supplementary Fig. 1). Indeed, for the groups closest to the reference phenotype (2-fold and 3-fold deviations), the engineered strains retained the dynamic characteristics of the reference strain while producing higher anthranilate titers (>15%)

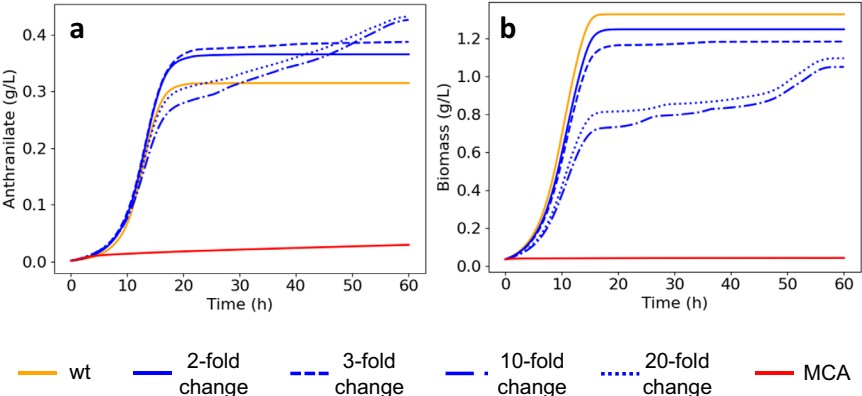

**Fig. 3 | Impact of limiting phenotype perturbation on resulting designs.** Mean anthranilate (**a**) and biomass (**b**) responses of engineered strains under different allowable fold changes in concentrations with respect to the reference strain (wt). As we permit a greater deviation from the reference physiology, from 2-fold (solid blue) to 20-fold (dotted blue) changes in concentrations, we observe a decrease in the titers of anthranilate and biomass across K_trpD9923 at the end of the fermentation period of the reference strain (18 h). Furthermore, the completely unconstrained approach to strain design that uses the top-3 control coefficients alone (MCA) stifles growth as well as anthranilate production. These results underline the importance of adhering to the reference physiology when conducting strain design. Source data are provided as a Source Data file.

at a modest cost to growth (<16%). Moreover, the titers achieved by these strains (~0.38 g/L) are only attained by the designs with 10-fold and 20-fold deviations after more than double (~40 h) their fermentation time.

In contrast, the designs with 10-fold and 20-fold deviations demonstrated slower dynamics than the reference strain, with lower mean titers and growth at the end of the production period for the latter (Fig. 3). In a similar vein, the unconstrained designs ('MCA' in Fig. 3) consistently failed to achieve any semblance of growth or production of anthranilate. The likely explanation for this is that, by not constraining the phenotype perturbation, we pushed the engineered strains far away from the reference strain while disregarding the network effects of the enzyme modifications. Even when we considered the targets that had a non-negative impact on growth, the resulting responses were inferior to the designs generated using NRA (Supplementary Note 2, Supplementary Figs 2–4).

These results suggest that it is judicious to generate designs that minimize phenotype perturbation while respecting other design specifications, such as maximal titer and specific productivity, for the cost-effective production of valuable biochemicals.

## NOMAD designs include validated experimental targets

We employed NRA[26] to engineer strains with a maximized yield of anthranilate with respect to glucose uptake for K_trpD9923 ("Methods" section). Using the results of the phenotype perturbation study, we permitted no more than a 3-fold change in concentrations to ensure we adhered to the phenotype of the reference strain. In addition, we allowed three enzyme modifications with a maximum of 5-fold upregulation and unrestricted downregulation. There were multiple designs for each model within 5% of the maximal increase in anthranilate yield. The number of such design alternatives for each model ranged from 2 to 12. In total, we obtained 70 designs involving 37 enzymes across the 10 models, predicting a 90–158% increase in anthranilate yield. Out of the 70 designs, 41 were unique by membership, meaning that they contained a unique set of three enzymes to be targeted. A clustering analysis of the 41 unique designs revealed five distinct groups of alternative enzymatic manipulations meeting the design specifications (Fig. 4 and Supplementary Note 3).

We found that 8 out of the 37 enzymes involved in the designs were validated experimental targets for increasing the flux through the shikimate pathway[41–43]. Three out of these 8 enzymes, DDPA, DHQS, and SHKK, belonged to the shikimate pathway, three belonged to glycolysis, PGI, PPS, and PYK, and two belonged to the pentose

phosphate pathway, TALA and G6PDH2r (Fig. 4). More specifically, Patnaik et al. increased the carbon flow through the shikimate pathway by overexpressing DDPA (*aroG*) alone and DDPA along with PPS (*ppsa*)[43]. All our designs recommend upregulating DDPA, and one, in particular, suggests overexpressing both DDPA and PPS, along with SHKK (*aroK*). Rodriguez et al. reviewed strategies that sought to increase the production of aromatic amino acids by either increasing the availability of the precursors to the shikimate pathway or enhancing the activity within the pathway[42]. The reviewed strategies directly targeting the shikimate pathway included the upregulation of DDPA, DHQS (*aroB*), or SHKK. Among the experimental strategies that targeted phosphoenolpyruvate (pep) or erythrose-4-phosphate (e4p) availability were those that inactivated the pyruvate kinases (*pykAF*), increased the activity of PPS, knocked out PGI (*pgi*), or redirected carbon to the pentose phosphate pathway (PPP) through the upregulation of TALA (*talB*), TKT (*tktA*), or G6PDH2r (*zwf*). NOMAD designs contained all previously mentioned interventions except for TKT, both in terms of the target enzyme and the direction of these manipulations (whether it was over or under-expression).

Interestingly, although one of the generated designs proposes PYK downregulation in line with the experimental approach (Fig. 4, DDPA↑, SHKK↑, PYK↓), another design proposes its upregulation instead (DDPA↑, PYK↑, GLUDy↓), suggesting the possibility of alternative regulation patterns when targeting multiple enzymes simultaneously.

In addition to encompassing several reported experimental interventions, NOMAD also suggested targets that can achieve the same impact on anthranilate production as the expert-proposed candidates. Some of them frequently appeared in our designs, such as the downregulation of GLUDy (11/41 designs) and the upregulation of GLNS (8/41 designs). In contrast, the upregulation of ENO and ICL appeared only in one design each.

## K_trpD9923 models qualitatively capture recombinant behavior

The experimental study by Balderas-Hernandez et al.[28] included fermentation data from two overproducing strains, W3110 *trpD9923*/pJL*aroG*fbr and W3110 *trpD9923*/pJL*aroG*fbr*tktA*. Both strains contained a feedback-resistant version of *aroG*, while the latter also included the overexpression of transketolase (*tktA*). The two strains achieved mean anthranilate titers of 0.44 g/L and 0.75 g/L, respectively.

We implemented the genetic modifications of these two strains in K_trpD9923 to evaluate the predictive performance of the kinetic models against the experimentally observed responses. The

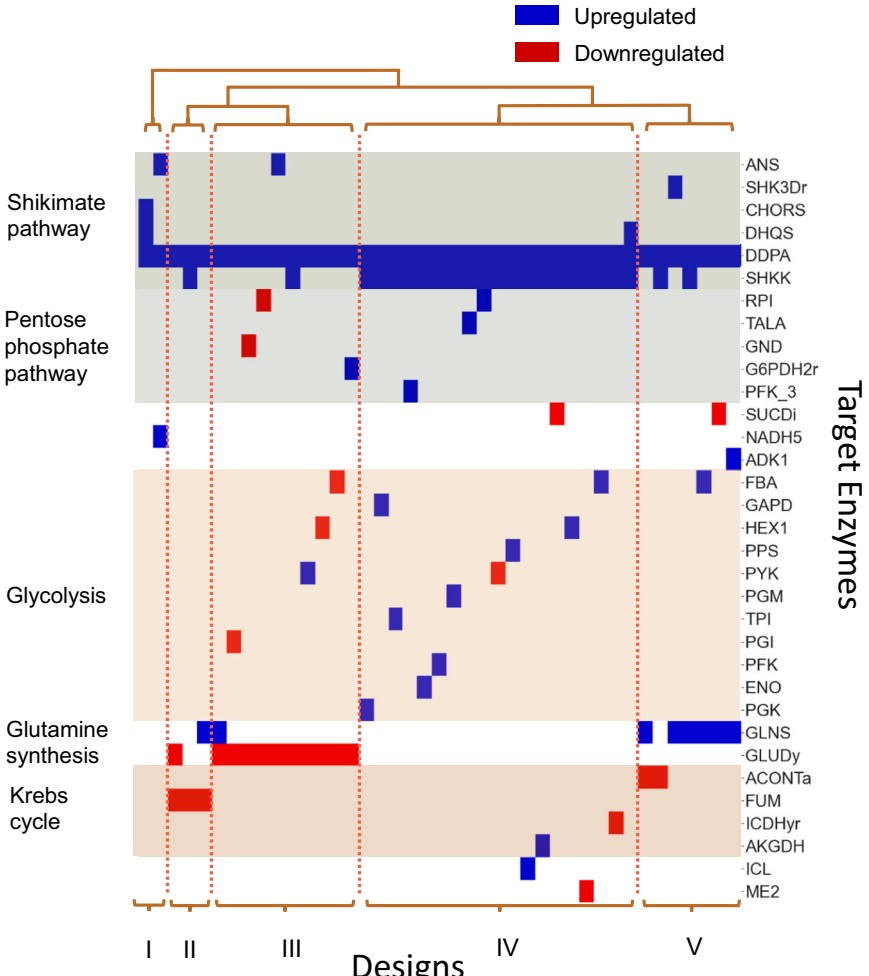

**Fig. 4 | Clustering analysis of the 41 unique NRA designs from K_trpD9923 reveals 5 distinct routes for overproducing anthranilate.** Each design contains a unique set of enzymes to be targeted. SHKK Shikimate kinase, CHORS Chorismate synthase, DHQS 3-dehydroquinate synthase, CHORM Chorismate mutase, ANS Anthranilate synthetase, ANPRT Anthranilate phosphoribosyltransferase, DDPA 3-deoxy-D-arabino-heptulosonate 7-phosphate synthetase, GND Phosphogluconate dehydrogenase, PFK_3 Phosphofructokinase (s7p), RPI Ribose-5-phosphate isomerase, G6PDH2r Glucose 6-phosphate dehydrogenase, NADH5 NADH dehydrogenase, SUCDi Succinate dehydrogenase (irreversible), HEX1 Hexokinase, PGK Phosphoglycerate kinase, PYK Pyruvate kinase, PGM Phosphoglycerate mutase, PGI Glucose-6-phosphate isomerase, PPS Phosphoenolpyruvate synthase, GAPD Glyceraldehyde-3-phosphate dehydrogenase, FBA Fructose-bisphosphate aldolase, PFK Phophofructokinase, TPI Triose-phosphate isomerase, FBP Fructose-bisphosphatase, ENO Enolase, PSERT Phosphoserine transaminase, GLUDy Glutamate dehydrogenase, GLNS Glutamine synthetase, ACONTa Aconitase, AKGDH 2-Oxogluterate dehydrogenase, ICDHyr Isocitrate dehydrogenase, FUM Fumarate mutase, ICL Isocitrate lyase, ME2 Malic enzyme (NADP). Source data are provided as a Source Data file.

experimental data were extracted from the work by Balderas-Hernandez et al. using an online tool, Webplotdigitizer[44]. To simulate the feedback-resistant version of aroG in W3110 *trpD9923*/pJL*aroG*fbr, we modified the K_trpD9923 models by removing DDPA inhibition by phenylalanine, creating K_trpD9923_d1. Similarly, to simulate W3110 *trpD9923*/pJL*aroG*fbr*tktA*, we removed DDPA inhibition and increased the enzyme activities of TKT1 and TKT2, creating the in-silico double mutant, K_trpD9923_d2. Although the *in-silico* versions of the two strains provided lower median titers of anthranilate than those reported experimentally, they captured the performance trends of both interventions: K_trpD9923_d2 produced better median titers (0.35 g/L) than K_trpD9923_d1 (0.33 g/L) which were, in turn, superior to K_trpD9923 (0.31 g/L) (Fig. 5). Our simulated strains adhered more closely to the glucose uptake and growth of the reference strain than those observed in the experimental strains (Supplementary Note 4, Supplementary Fig. 5). This suggests that the in-silico strains diverted less carbon toward the shikimate pathway than the experimental strains did. Upon investigation of K_trpD9923_d2, we found that we needed to overexpress additional enzymes in the shikimate pathway to

achieve titers that matched those of the experimental double mutant (Supplementary Note 5, Supplementary Fig. 6). Interestingly, although we did not integrate information about the two recombinant strains in the model-building process, our models could reproduce the experimental observation that the difference in anthranilate titers between the two engineered strains was more significant than the difference between the wild-type and W3110 *trpD9923*/pJL*aroG*fbr strain.

We then evaluated the performance of the NOMAD designs by comparing their performance in fermentation simulations against those of K_trpD9923_d1 and K_trpD9923_d2. The superior median anthranilate titers attained by the NOMAD designs (0.42 g/L) compared to the *in-silico* versions of the two strains suggested that the strains W3110 *trpD9923*/pJL*aroG*fbr and W3110 *trpD9923*/pJL*aroG*fbrtktA could be further improved (Fig. 5).

**Prioritizing NOMAD designs for robust implementation**
We cannot know, a priori, which kinetic model from a population of models most accurately represents a cell's physiology. We thus conducted a two-stage screening process to ensure that the engineering

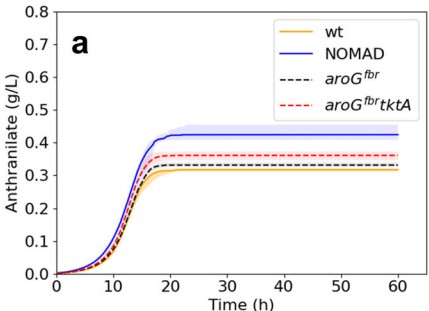
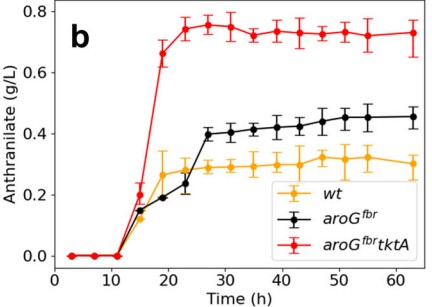

**Fig. 5 | Performance of K_trpD9923 models implementing modifications of recombinant strains. a** Simulated responses of W3110 *trpD9923* (wt, orange), W3110 *trpD9923*/pJL*aroG*fbr (black) and W3110 *trpD9923*/pJL*aroG*fbr*tktA* (red) when they are implemented in K_trpD9923. For comparison, we provide the responses of the implemented NOMAD designs (blue). The bold lines and shaded regions represent the median and interquartile ranges of the responses across the 10 kinetic models. **b** Experimentally recorded data for the three strains. The solid circles and error bars represent the mean and standard deviations of the triplicate experiments. The models capture the experimentally observed trends, with the overexpression of *tktA* resulting in a superior anthranilate titer compared to the targeting of *aroG* alone. NOMAD designs provide a superior titer of anthranilate when compared with the in-silico implementation of the experimental designs. The temporal evolutions of extracellular glucose and biomass of the in-silico strains displayed in panel are provided in Supplementary Note 4. Source data are provided as a Source Data file.

strategies we proposed for experimental implementation remain consistent across the range of phenotypes these models represent. In the first stage, we selected designs based on their expected performance across all the K_trpD9923 models NRA. In the second stage, we subjected these designs to more rigorous performance and sensitivity tests in a bioreactor setting. The preliminary screening, using log-linear approximations, mitigates the computational cost associated with nonlinear simulations.

We evaluated the 41 unique designs for each of the models in K_trpD9923 using NRA, ranked them, and selected the top five designs with the highest mean predicted increase in anthranilate yield (~93%) for further scrutiny (Supplementary Note 6, Supplementary Fig. 7, Supplementary Table 1). Interestingly, all these designs belong to Cluster III in Fig. 4 and suggest redirecting carbon to the shikimate pathway by upregulating DDPA and increasing glutamate availability for glutamine synthesis by downregulating GLUDy (Supplementary Note 6, Supplementary Fig. 7). Four of the designs also balance the availability of pep and e4p by targeting glycolysis (HEX1, PYK, PGI) or the pentose phosphate pathway (GND). The fifth design increases the enzyme activity of ANS, which is responsible for anthranilate synthesis.

We then subjected these five designs to nonlinear batch fermentation simulations that closely mimic real-world conditions ("Methods" section). Based on the evaluation results, we retained four designs that balance the availability of pep and e4p (Supplementary Note 6, Supplementary Fig. 8). Due to the consistent presence of DDPA and GLUDy in all top designs, we evaluated the performance of designs when perturbing only these two enzymes. The results showed comparable anthranilate titers to the triple mutants, highlighting their significance in redirecting carbon toward anthranilate production (Supplementary Note 7, Supplementary Fig. 9). The results of the in-silico validation studies underline one of the key features of NOMAD: it is only through the use of nonlinear simulations that we could conduct such quality checks and glean insights into the applicability of the different designs.

**Multi-strain model calibration improves quantitative accuracy**
While the K_trpD9923 models successfully captured the qualitative trends of the experimental strains W3110 *trpD9923*/pJL*aroG*fbr and W3110 *trpD9923*/pJL*aroG*fbr*tktA*, they fell short in quantitatively reproducing their experimental titers (Fig. 5). To enhance the quantitative accuracy of our models and provide refined designs for overproducing anthranilate, we developed kinetic models by integrating additional data on the physiology of recombinant strains.

We first generated 4,000,000 putative kinetic models using ORACLE and filtered them based on their linearized dynamics ("Methods" section). We then screened the remaining models based on their ability to match the experimentally observed growth, glucose uptake, anthranilate titer, and dynamics of W3110 *trpD9923*, as well as the anthranilate titer of W3110 *trpD9923*/pJL*aroG*fbr ("Methods" section). The screening yielded 35 models that could accurately reproduce the behavior of these two strains (Fig. 6a). The 35 models also outperformed K_trpD9923 in reproducing the anthranilate titer of W3110 *trpD9923*/pJL*aroG*fbr*tktA*, achieving median titers of 0.65 g/L. This value is significantly closer to the experimentally obtained titers of 0.75 g/L compared to K_trpD9923_d2, which predicted a titer of 0.35 g/L. These results demonstrate the benefit of integrating additional information into the models when accessible.

Among these 35 models, 13 accurately reproduced the experimentally observed anthranilate titers for all three strains (Fig. 6b). We refer to these 13 models as eK_trpD9923, where 'e' stands for enhanced. The in-silico implementations of W3110 *trpD9923*/pJL*aroG*fbr and W3110 *trpD9923*/pJL*aroG*fbr*tktA* strains are labeled as eK_trpD9923_d1 and eK_trpD9923_d2, respectively.

**Designs from enhanced models provide superior performance**
We used eK_trpD9923 and eK_trpD9923_d2 to generate designs that produced at least 95% of the maximal anthranilate yield for each model ("Methods" section). For eK_trpD9923, there were 123 such designs involving 34 unique enzyme targets. Bioreactor simulations of these 123 designs showed a higher median anthranilate titer (0.78 g/L) than the two experimental strains (Fig. 7a).

For eK_trpD9923_d2, 36 designs encompassed 14 different enzymes, with 13 designs being unique by membership. The decrease in the number of unique designs compared to the designs from eK_trpD9923 suggests that there were few remaining ways to augment the performance of the double mutant. We implemented these designs in bioreactor simulations and found that the mean anthranilate titer (1.16 g/L) was significantly superior to that attained by eK_trpD9923_d2 (0.71 g/L) (Fig. 7b). The designs also had a 6% increase in productivity (0.032 g/L/h vs. 0.03 g/L/h) when compared to eK_trpD9923_d2 despite reaching 90% of their maximal titers at a mean of 33 h as opposed to 21 h.

We performed a clustering analysis on the designs devised with eK_trpD9923 and eK_trpD9923_d2, which unveiled two distinct groups of designs sharing common characteristics in the former and three such groups in the latter (Supplementary Note 8, Supplementary Figs 10 and 11).

Following this, we conducted a two-stage design screening, as outlined in the "Methods" section. We ultimately selected the top 5

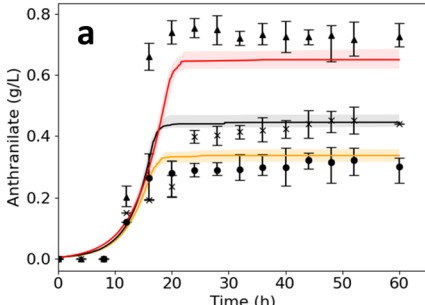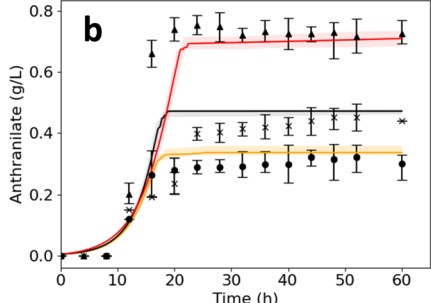

**Fig. 6 | Impact of multi-strain data integration on model accuracy.** Anthranilate titers from **a** the 35 models calibrated on data from two strains – W3110 *trpD9923* and W3110 *trpD9923*/pJL*aroG*fbr, and **b** the 13 models calibrated on all three strains, including W3110 *trpD9923*/pJL*aroG*fbr*tktA*. Simulation results are denoted by solid lines (median) and shaded regions (interquartile range) for W3110 *trpD9923* (orange), W3110 *trpD9923*/pJL*aroG*fbr (black), and W3110 *trpD9923*/pJL*aroG*fbr*tktA*

(red). Results of the triplicate experiments are denoted by solid circles (mean) and error bars (standard deviation) for W3110 *trpD9923* (circle), W3110 *trpD9923*/ pJL*aroG*fbr (cross), and W3110 *trpD9923*/pJL*aroG*fbr*tktA* (triangle). Incorporating additional data improves accuracy in capturing the experimental behavior of all three strains. Source data are provided as a Source Data file.

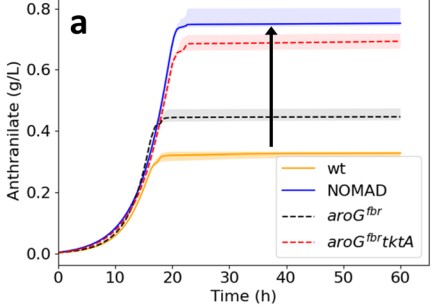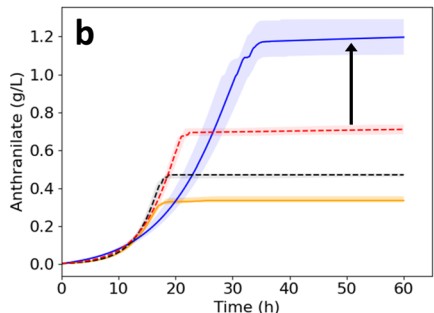

**Fig. 7 | Performance of NOMAD designs implemented in enhanced models.** **a** Predicted temporal evolution of anthranilate titers of the NOMAD designs (blue) generated using enhanced models of the reference strain - eK_trpD9923. The solid lines and shaded regions denote the median and interquartile ranges, respectively. The designs attained a median titer (0.78 g/L) that was higher than those of the reference strain - *trpD9923* (orange), and the in silico implementation of the two

experimentally implemented strains, W3110 *trpD9923*/pJL*aroG*fbr (black) and W3110 *trpD9923*/pJL*aroG*fbr*tktA* (red). **b** Anthranilate curves of the designs for improving the performance of the in-silico implementation of W3110 *trpD9923*/pJL*aroG*fbr*tktA* - eK_trpD9923_d2. These designs attained a median titer of 1.16 g/L, superior to the simulated titers of the three experimental strains and the NOMAD designs from eK_trpD9923. Source data are provided as a Source Data file.

designs as the most robust candidates for experimental implementation (Supplementary Note 9, Supplementary Figs 12 and 13, Supplementary Table 2). In the case of eK_trpD9923, all five designs recommended the upregulation of ANS and DDPA, differing primarily in their choice of the third enzyme, which included FBA, GAPD, PGK, G6PDH2r, and PGL. These five designs also outperformed the designs devised using K_trpD9923 when the latter were implemented in eK_trpD9923 (Supplementary Note 10, Supplementary Fig. 14). For eK_trpD9923_d2, the top 5 designs all featured ANS upregulation, coupled with one of the following combinations of interventions: DHQS and GLUDy, DHQS and AKGDH, DDPA and GAPD, DDPA and FBA, or DDPA and PGL. Expert knowledge can be employed to conduct a comparative analysis of the suggested designs and further refine the selection of designs for experimental implementation to enhance the performance of strains W3110 *trpD9923* and W3110 *trpD9923*/ pJL*aroG*fbr*tktA*.

## Discussion

Rational strain design using kinetic models is one of the holy grails of metabolic engineering since it obviates the need for expensive, high-throughput experiments and provides a structured approach to strain design. NOMAD provides a systematic framework to achieve this by using a first-principles-led approach to build quality kinetic models and then conduct rational strain design using a judicious choice of design specifications and constraints.

Although several frameworks exist to produce kinetic models that are representative of *steady-state* behavior, we demonstrate that

carefully choosing models based on their *dynamics* is also essential. Through our multi-step screening process, we use fundamental engineering principles to obtain high-quality kinetic models in the first study that not only reproduce the dynamics of the reference strain but also capture the trends observed while implementing experimental engineering strategies.

The usual approach for acquiring a collection of high-quality kinetic models involves creating a large pool of candidate models and then pruning them based on several established criteria. However, this process can be quite burdensome. Alternatively, one could aim to improve the incidence of high-quality models during the generation phase itself. We have developed several machine-learning techniques to achieve this goal, including iSCHRUNK[20,22], REKINDLE[21], and RENAISSANCE[31]. The last two methods have proven particularly effective by using stratified sampling to produce high-quality models within kinetic spaces that satisfy the aforementioned criteria.

Using the first case study, we have also shown that it is strongly recommended to maintain the phenotype of the engineered strains close to the reference strain. Because we have access to efficient, accurate, and well-established tools for predicting metabolic responses for small perturbations, such as Network Response Analysis[26], we retain prediction accuracy without too much perturbing the physiology. In this manner, NOMAD overcomes the limited accuracy posed by log-linear approximations by using smaller, more constrained steps that provide more robust results in a nonlinear setting. Such a philosophy of cautiously improving the system performance is similar to the windsurfer approach[45] in systems and control theory. In the broader

context of design-build-test-learn cycles, it might be advantageous to make incremental but reliable improvements, use experimental data to calibrate models further, and then conduct another computational study to design subsequent strains.

It is tempting to assume that the strain design process with high-quality models is seamless. On the contrary, we have shown that avoiding the inherent combinatorial explosion when conducting unguided and unbiased rational strain design is not trivial. We achieve this by judiciously framing an optimization problem around control coefficients to enumerate multiple routes to achieve the design objective efficiently. In this manner, NOMAD balances between accuracy (offered by nonlinear simulations) and computational efficiency (associated with log-linear kinetics) and provides a scalable and computationally efficient way to constrain the network effects of the proposed changes while achieving the desired metabolic objective. Although NOMAD ranks the designs using a defined set of quantitative criteria, experts working on the problem often possess valuable biological expertise that allows them to prioritize among the top solutions. A mathematical model may not readily capture their insights, knowledge, and experience. Nevertheless, NOMAD offers a limited number of suggestions that satisfy constraints that are not easy to account for by conceptual design alone.

The enhanced models developed in the second case study effectively reproduced anthranilate production in all three experimental strains. Moreover, they enabled us to propose reliable targets for further improving the double mutant strain. This study demonstrated that the quality of the models significantly improves with the integration of data from multiple physiologies. It also underscored the importance of extensive datasets for developing highly accurate metabolic models. Indeed. even for sophisticated modeling techniques, such datasets are essential for unraveling the complex, nonlinear nature of metabolic responses.

Overall, NOMAD presents a versatile, modular framework whose concepts are applicable regardless of the model size, the type of kinetic mechanisms used, or the framework used to build the putative models. In doing so, it paves the way for accelerating the use of kinetic models in strain design endeavors.

## Methods

### Generating kinetic models of *E. coli* strain W3110 *trpD9923*

As a proof of concept of the NOMAD framework, we use kinetic models to propose rational design strategies for the overproduction of anthranilate in *E. coli* W3110 *trpD9923*. This strain accumulates anthranilate due to a loss of anthranilate phosphoribosyltransferase (ANPRT) activity leading to tryptophan auxotrophy. We require knowledge of the reaction mechanisms and the parameters that characterize each mechanism to build such kinetic models. The complexity of the metabolic network coupled with physiological and parametric uncertainty, renders this a challenging task. We used the ORACLE framework[15,16,29,36,46] to overcome these challenges and develop a set of putative kinetic models representing the strain for two case studies. In the first study, we developed kinetic models K_trpD9923 that were calibrated using the data acquired on the reference strain alone. In the second study, we developed kinetic models eK_trpD9923 that were calibrated using data derived from the experimentally observed behavior of the reference and two recombinant strains. We generated the models through the following steps:

1. Reduced model generation: To build a kinetic model, we first need stoichiometric information about the metabolic network. We used redGEM[47] and lumpGEM[48] to create a reduced model of *E. coli* and then removed the ANPRT reaction to mimic the nonsense mutation in *trpD9923*. We retained all reactions belonging to the core subsystems – glycolysis, pentose phosphate pathway (PPP), the Krebs cycle, anaplerotic reactions, the shikimate pathway, and glutamine synthesis, and added a single reaction for

growth by lumping the biosynthetic reactions. The resulting network had 196 reactions (with 81 transport reactions) and 159 metabolites, spread across 2 compartments, the cytoplasm and the periplasm.

2. Data integration – metabolomics, fluxomics & thermodynamics: Before generating samples of steady-state concentrations and fluxes for the two studies, we integrated exo-metabolomic and exo-fluxomic information obtained at the start of the exponential phase for the reference W3110 *trpD9223* strain[28]. Since no lag-phase was observed in this strain, this corresponded to the start of the fermentation process itself. For the glucose uptake rate and the growth rate, we fitted analytical batch fermentation curves to the experimental data[28]. We used information on the M9 minimal medium content to constrain the extracellular metabolite concentrations. In addition to this, we integrated general metabolomics and thermodynamics data[49–51].

   In the second study, our primary objective was to enhance the quantitative accuracy of our predictions. To achieve this, we constructed improved kinetic models by integrating additional information derived from the experimental strains. Before embarking on this endeavor, we sought to comprehend why the models in the first study failed to capture the titers for W3110 *trpD9923*/pJL*aroG*fbr*tktA* accurately. In experiments, this strain produced anthranilate titers (0.75 g/L) that were 150% and 75% higher than W3110 *trpD9923* (0.31 g/L) and W3110 *trpD9923*/pJL*aroG*fbr (0.44 g/L), respectively, indicating that transketolase activity had a significant impact on anthranilate production. In comparison, the kinetic models of the strain, K_trpD9923_d2, were relatively insensitive to tktA overexpression as evinced by their reduced titers (0.35 g/L). Our analysis revealed the root cause of this poor response is that TKT1 and TKT2 are close to thermodynamic equilibrium. It is well known in the literature that enzymes that operate near equilibrium do not exert significant control over metabolic fluxes[18]. We incorporated this insight when building models in the second study and ensured that TKT2 was displaced away from equilibrium by forcing its Gibbs free energy to be at least 1.25 kcal/mol.

3. Sampling of steady-state concentrations and fluxes: We used thermodynamics-based flux balance analysis (TFA)[52] implemented in pyTFA[53] to generate 4000 steady-state samples that resulted in at least 80% of the maximal growth for both studies. These samples contain fluxes, concentrations, and thermodynamic variables associated with each reaction ($\Delta G'^0$, $\Delta G'$). Incorporating thermodynamic information in TFA guarantees that the sampled fluxes and concentrations adhere to the second law of thermodynamics.

4. Data integration – kinetic reaction mechanisms: Depending on the stoichiometry of each reaction in the metabolic network, we assigned a reaction mechanism (Supplementary Note 11, Supplementary Data 1). The primary mechanisms we used were the Generalized Reversible Hill[54], and Convenience kinetics[55], both of which capture enzyme saturation. We used mass action kinetics to model periplasm to extracellular transports. Considering the importance of regulatory networks within the cell, we also modeled four types of allosteric regulation: (i) competitive inhibition, (ii) uncompetitive inhibition, (iii) mixed inhibition, and (iv) activation. We obtained regulatory information from an earlier kinetic modeling study[56]. Considering the focus of our work on anthranilate production through the shikimate pathway, we also added the inhibition of aroG by phenylalanine[57] and the inhibition of ANS by tryptophan[58]. Overall, we incorporated regulatory information for 31 reactions, including interactions for 5 reactions in the Shikimate pathway (Supplementary Note 11, Supplementary Data 2).

5. Kinetic model generation: With the stoichiometry and reaction mechanisms at hand, we needed to determine the kinetic

parameter sets that characterize the system of ODEs using the ORACLE framework[15]. We built putative kinetic models for the first case study by generating sets of kinetic parameters around each of the 4000 steady-state samples. ORACLE ensures that the sampled kinetic parameters are consistent with the steady-state concentrations and the thermodynamic displacements of each reaction. Each combination of a steady-state profile and its associated kinetic parameter set constituted one kinetic model. During sampling, we pruned the kinetic models for linear stability and retained 200 models per steady state whose Jacobian matrix had all negative eigenvalues. In this manner, we generated 800,000 kinetic models that were locally stable around the reference steady states.

For the second study, we generated 1000 sets of kinetic parameters for each steady state, yielding a total of 4,000,000 locally stable models. We sampled more models in the second than in the first study to increase the probability of finding models that could reproduce the behavior of all three strains in a batch fermentation setting.

## Screening kinetic models

Once the initial set of kinetic models is available, we screen them to find the ones that are representative of the dynamic characteristics of the reference strain. The screening process follows four consecutive criteria to evaluate the models: (i) physiologically relevant linearized dynamics, (ii) robustness to perturbations in a nonlinear context, (iii) ability to reproduce experimentally observed temporal data, and (iv) robustness to enzymatic perturbations. In each step, we increase the robustness and quality of the models that satisfy the requisite criteria. More specifically, during the screening process, we screen for:

1. Linearized dynamics: We built each kinetic model from the initial set around a steady-state consistent with the integrated experimental data. However, not all of these models necessarily capture the experimentally observed dynamics of the metabolic network. To identify models with physiologically relevant dynamic properties, we assume that: (i) any experimentally observable steady state is locally stable; and (ii) since metabolic reactions occur at a timescale of seconds and milliseconds, metabolic processes should settle before the cell division, which is at a timescale of minutes and hours.

   To this end, we first linearize the models around their steady states, and estimate the time constants using the eigenvalues of the Jacobian. To compute the Jacobian, we need the kinetic parameters computed by a kinetic modeling technique and the steady-state concentrations in the metabolic network. These concentrations can be obtained by integrating the set of ODEs till they reach a steady state as done by MASSpy[17] or Ensemble Modeling (EM)[19], or directly from the constraint-based models used to build the kinetic models as in pyTFA[53]. We then use the calculated eigenvalues and time constants to screen the models. A necessary condition for local linear stability of the system around the steady state is that its Jacobian has eigenvalues with negative real parts, i.e., Re $(\lambda_i) < 0, \forall i$, implying that all infinitesimal perturbations to the system will eventually return to the original state of the system. However, our interest extends beyond whether the perturbed system will return to its original state but also how quickly it will do so. Therefore, in addition to screening based on local stability, we also use the dominant time constant of the model, $\tau_d$ (Eq. 1), to screen for models with physiologically relevant dynamics.

$$\tau_d = \frac{1}{\min_i |\text{Re}(\lambda_i)|} \qquad (1)$$

Metabolic processes occur on different time scales, ranging from milliseconds to minutes, and the dominant time constants depend heavily on the organism. Metabolic responses to perturbations should settle down before cell division. According to the dynamical systems theory, these responses should reach 99.3% of their original steady states within 5 dominant time constants. Therefore, to have a model whose responses settle by the doubling time, its dominant time constants should be five times smaller than the doubling time.

The reference strain, *E. coli* W3110 *trpD9923*, had a maximal in-silico growth rate of 0.32/h, corresponding to a doubling time of 130 min. For building K_trpD9923 in the first study, we chose kinetic models with a dominant time constant of less than 24 min. Models that meet this condition will likely return to 99.5% of their steady-state value within the doubling time when subjected to small perturbations (Supplementary Note 12). We set the time constant threshold for the enhanced models, eK_trpD9923, at 43 min, indicating that models meeting this criterion would return to 95% of their steady-state values within the doubling time. This relaxed constraint expanded the selection of models available for matching the experimentally observed behavior of the three strains.

2. Nonlinear response to concentration perturbations: The preceding linear stability analysis provides information on how the network will respond to *infinitesimally small* perturbations. However, in actual fermentation settings, the cells traverse different phases, such as the lag and exponential phases, during which there are significant fluctuations in concentration profiles. Before using the chosen kinetic models for strain design, we want them to be robust to such large-scale concentration changes. In the presence of experimental fermentation data, we can directly verify the robustness of our models by checking if they can reproduce the fermentation curves. However, in the absence of such temporal data, we can verify the robustness of the models using their nonlinear responses to randomly applied concentration perturbations. To do this, we apply a 'k-fold' perturbation to the steady-state concentrations of each kinetic model and integrate the system of ODEs to verify whether or not the perturbations are damped out before the cell's doubling time. We repeat this 'n-times' and select those models for which all the perturbed models return to the original steady state within the physiological timescale of the cell.

   This step was unnecessary for both studies as we had ample fermentation data to calibrate our models.

3. Reproduction of batch fermentation data: NOMAD integrates information on the temporal evolution of key metabolite concentrations, when available, by selecting models that reproduce the data within reasonable bounds. We first integrate all known information about the inoculum and the fermentation medium and run batch fermentation simulations using the models that are selected in the previous step. We then choose those models that can accurately capture experimental fermentation data, which is available in the form of growth curves, secretions, and uptakes. In the first study, we calibrated models to reproduce the behavior of the reference strain, W3110 *trpD9923*, in batch fermentation experiments. We integrated inoculum information provided in the experimental work[28] and ran nonlinear simulations for each model that passed the screening based on linearized dynamics. We then chose those models within 5% and 10% of the final steady-state values of growth and extracellular anthranilate, respectively, whose fermentation times were less than 20 h. Including fermentation time in the pruning ensured that the models adhered to the experimentally observed *dynamics* of the strain in addition to the final titers.

The kinetic models in the second study were calibrated on data from the reference strain, W3110 trpD9923, and from the two overproducing strains, W3110 *trpD9923*/pJL*aroG*<sup>fbr</sup> and W3110 *trpD9923*/pJL*aroG*<sup>fbr</sup>*tktA*. We integrated inoculum and medium data and simulated all three strains using the models that passed the screening based on linearized dynamics. For the two recombinant strains, we removed the inhibition of DDPA by phenylalanine by setting the inhibition constants to arbitrarily large numbers. Furthermore, we modeled tktA overexpression by applying a 10-fold increase to the activities of TKT1 and TKT2. We first selected models that could reproduce the behavior of W3110 *trpD9923* and W3110 *trpD9923*/pJL*aroG*<sup>fbr</sup> using the following criteria: (i) the simulated final growth and anthranilate titers of the reference strain should be within 10% and 20% of their experimental values (1.29 g/L and 0.31 g/L), (ii) the simulated anthranilate titers of the recombinant strains should be within 12.5% of those recorded experimentally (0.44 g/L). The models that passed this screening were further filtered for those that could produce anthranilate titers that were at least 90% of the experimentally reported values for W3110 *trpD9923*/pJL*aroG*<sup>fbr</sup>*tktA* (0.75 g/L) when the double mutant was implemented in-silico.

4. Robustness to enzymatic interventions: The end goal of the framework is to provide targets for enzymatic interventions that enable us to achieve a given metabolic output. Not all the models that are selected in the previous screening steps are equally robust to enzymatic interventions - some can veer significantly from their behavior, showing little to no growth, while others can retain their reference growth level. Hence, to determine the robustness of each kinetic model to such interventions we apply a 'k-fold' perturbation to the maximal velocities of the different reactions and study the growth of the resulting strain. We repeat this 'n' times and choose those models for which all the perturbed strains demonstrate satisfactory growth. This final screening step ensures that the models are not only representative of the reference strain but also suitable for enzymatic interventions.

When building models for the first study, we applied a 10% normally distributed perturbation to the maximal velocities of each reaction in the network and then integrated the system of ODEs in a batch reactor setting. We repeated this process 50 times for each kinetic model and chose those models that displayed at least 50% of the experimentally observed biomass for all 50 perturbations.

We skipped the screening based on random perturbations for the kinetic models in the second study, eK_trpD9923, as they had already proved their ability to withstand different engineering interventions.

This multi-step filtering process produces a *population* of robust, representative kinetic models that are adequate for rational strain design.

## Robust strain design using kinetic models

We use the screened kinetic models to conduct rational strain design with a given objective to be attained. For both case studies, the objective was to maximize the increase in yield of anthranilate with respect to glucose uptake. The strain design process can be divided into the following steps:

1. Generating design alternatives using Network Response Analysis: One approach to strain design would be to exhaustively simulate all possible combinations of target enzymes along with the degrees of up or down regulations applied to them. The arduous nature of this task and the computational cost involved provide a strong case for a more judicious approach to choosing enzymatic targets.

   One possibility would be to use Metabolic Control Analysis (MCA)[39,40,59], i.e., to calculate the log-linear sensitivities of the

production pathway to system parameters and to then use the enzymes with the top control coefficients as the candidates to be tested in a nonlinear setting. However, this approach has its drawbacks. An increase in enzyme activity affects not only the target flux/metabolite but also other components of the network, potentially causing a significant deviation from the reference physiology, or the accumulation of toxic metabolites. This situation is further complicated when targeting multiple enzymes simultaneously. A starting point to overcome such deleterious effects would be to use heuristics and expert knowledge and eliminate from contention those targets that are known to have undesirable network effects.

To provide a more systematic and efficient approach to dictate such choices and constraints, a constraint-based MCA method called Network Response Analysis (NRA) was developed[26]. In NRA, we frame the strain design objective as a mixed-integer linear optimization problem built around the control coefficients, and the reference steady-state profiles of concentrations and fluxes. In addition, we supply design constraints such as the allowable fold-change in fluxes, concentrations, and enzyme activities, and the number of allowable enzymatic interventions. In this way, NRA provides two distinct advantages. First, it ensures the reliability and robustness of designs by controlling the deviation from the reference phenotype through the imposed constraints. Second, by using an optimization problem, NRA provides a computationally efficient and scalable approach to strain design by avoiding the combinatorial explosion inherent when we seek multiple enzymatic targets.

With these features in mind, we use NRA to enumerate designs for each of the chosen kinetic models that achieve the desired objective within a certain threshold. In this manner, we can generate hundreds of designs across all the kinetic models. The detailed formulation of the NRA problem is available in Supplementary Note 13, Supplementary Tables 3–5.

We generated designs for the models in the first study, K_trpD9923, under the following constraints: (i) a maximum of 3-fold change in concentrations and 5-fold change in enzyme activities, (ii) a maximum of 3 enzymatic interventions, and (iii) a maximum of 20% reduction in growth rate. We set the objective to maximize the increase in anthranilate yield with respect to glucose uptake, with respect to that of the reference strain, and enumerated all designs within 5% of the maximal objective for each kinetic model.

In the second study, except for two changes, we retained the aforementioned parameters when generating designs using eK_trpD9923 and eK_trpD9923_d2. First, we increased the allowable fold changes in enzyme activity to 10; this corresponds to the fold changes that we applied to the maximal velocities of TKT1 and TKT2 while constructing eK_trpD9923_d2. Second, we compensated for the relaxed constraint on enzyme activities by reducing the permissible fold changes in concentrations to 2.5. This combination ensured that we did not perturb the system too far from its reference state while generating designs using NRA. For eK_trpD9923_d2, we first simulated the double mutant and recorded the steady-state concentrations and fluxes nine hours into fermentation when the strain was mid-way through the exponential growth phase. We then calculated the control coefficients at this steady state before using them to generate designs through NRA.

2. Two-stage ranking of designs: The product of the previous step is an aggregate of putative designs generated using each kinetic model. Designs generated using one model need not necessarily perform well when applied to other models. Therefore, we rank designs by the robustness of their performance across the phenotypic variation covered by the population of models. The

ranking is performed in two stages. First, we rank the designs based on their NRA-predicted performance across all models. Next, we assess the performance and robustness of the top-ranked designs in nonlinear simulations. The designs that fare well in this second stage are then proposed for experimental implementation.

Robustness can be characterized in many ways – the reappearance of the same design by membership across different models, or the highest predicted objective when the design is enforced across different models, etc. The criteria to define robustness can vary with the objective that we seek to attain.

For both studies in the presented work, we first extracted those designs that were unique by membership and used log-linear approximations of the kinetic models in NRA to assess their robustness across the models. We implemented each design in every kinetic model by setting the minimal log fold change in the activity levels of the associated enzymes to be 1e-6. We then maximized the increase in anthranilate yield with respect to glucose under the same constraints as those used to generate all the putative designs. Finally, we chose the 5 designs with the highest mean objective value across the models for further analysis.

After ranking and selecting the most robust designs in the first stage, we proceed to verify them in silico using a batch fermentation setup. We also investigate how sensitive the designs are to any uncertainties that may arise during experimental implementation. It is imperative to carry out this step since the designs' performance in the previous stage was evaluated using a log-linear approximation of the system, which was only taken at the reference steady state. By verifying the proposed designs in a nonlinear setup, we can gain a better understanding of how well the log-linear approximations perform in a nonlinear environment. Once we complete the design verification and analysis, we identify the most promising designs for experimental implementation.

For both case studies, we first analyzed the performance of the top 5 designs in a batch fermentation setting using the NRA-predicted enzyme fold changes specific to each model. For the inoculum and medium, we integrated the same information as was done in the model screening step. For the sensitivity analysis, we first calculated the mean NRA-suggested fold changes for each enzyme for a given design. We then applied a ±50% uniformly distributed perturbation to the mean fold changes for (i) all 3 enzymes, (ii) each enzyme individually while keeping the other two enzymes at the mean NRA-suggested fold changes. To obtain a statistical estimate of the sensitivities, we did this 10 times for each of the 10 models and tracked the mean of the 100 responses for each design.

### Reporting summary
Further information on research design is available in the Nature Portfolio Reporting Summary linked to this article.

## Data availability
Data extracted from the experimental work by Balderas-Hernandez et al.[28] are available in the GitHub [https://github.com/EPFL-LCSB/NOMAD]. Source data are provided with this paper.

## Code availability
The code was implemented in Python 3.6. The commercial solver IBM ILOG CPLEX Optimizer was used to solve the MILP problems. The code uses the SKiMPy and pyTFA packages, which rely on SUNDIALs and COBRApy, respectively. The code and data required to reproduce results are publicly available at GitHub [https://github.com/EPFL-LCSB/NOMAD]. A snapshot of the code, providing the exact version used in this study, is available through Zenodo [https://zenodo.org/records/10362313][60].

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

## Acknowledgements

This work was supported by funding from the European Union's Horizon 2020 research and innovation programme under grant agreement 814408 (B.N. and D.W.), the Swiss National Science Foundation Synergia grant CRSII5_198543 (M.M.), the Swedish Research Council Vetenskapsradet grant 2016-06160 (B.N.), and the Ecole Polytechnique Fédérale de Lausanne (EPFL).

## Author contributions

B.N., D.W., L.M. and V.H. conceptualized the study. B.N. and D.W. developed the software and performed the simulations. M.M. created the reduced stoichiometric model of E. coli. B.N. L.M., and V.H. analyzed the results and provided the discussion. B.N., L.M. and V.H. wrote the manuscript. L.M. and V.H. supervised the project and acquired the funding and the resources.

## Competing interests

The authors declare no competing interests.
