## [Peer Review File · Nature Communications]

Reviewers' Comments:

Reviewer #1:

Remarks to the Author:

In this work by Narayanan and coworkers they report a new computational framework for strain design called NOMAD. This method makes use of the advantages of kinetic models for metabolic engineering, such as capturing dynamic properties of the system and considering enzyme and metabolite concentration changes, however it reduces the search space to simplify the amount of possible solutions. The framework is novel and provides the basis for a design principle when using kinetic models: to seek the minimal deviation from the wild type phenotype. The presented data support the conclusions. They demonstrate how this design principle holds, then I consider that this could be a resource with great value to the design community and I can see how it will speed the DBTL cycle.

Major comments:

The main downside of this work is that it still requires enormous effort from the user to be implemented, such as but not limited to: it make use of previous methods (such as many kinetic model builds, NRA, MOMA etc.), It relies on the generation of thousands of kinetic models, great amounts of physiological and omic data, manual (or expert driven) quality checks and so on. Then it generates many possible solutions, and it is difficult to prioritize them. It seems that it will still be very difficult to be adopted by the average user.

In this case they begin with a large population of kinetic models "with experimentally observed omics and cultivation data, physicochemical laws, network topology, and regulatory interactions." What is the perspective to use this method in non typical growth conditions or non-model organisms?

The experimental validation in this case is fortunate in the shikimate pathway has been extensively explored for the production of several compounds so there are many references to show how the many combination of enzyme down and upregulation has been or not successful, however an average user of this method should prioritize among possible designs with out such information?

Can the authors provide another use case? and would it be desirable to provide a straightforward method to rank the provided designs so the experimental validation could start with the best cases?

Minor:

Abstract: where the six orders of magnitude come from?

Methods: Briefly define why it is important to retain only the models whose jacobian matrix had all negative eigenvalues

Methods: the steps to screen the kinetic models are not clear, it seems that they only describe the linearized dynamics

Reviewer #2:

Remarks to the Author:

In this work, the authors develop a workflow called NOMAD (NONlinear dynamic Model Assisted rational metabolic Engineering) to explore the design space within a kinetic model of metabolism. A major focus of this workflow is maintaining the physiological state near a healthy one (growth rate),

practically done by maintaining fluxes and metabolite concentrations near the reference state. They perform this analysis for the production of anthranilate, a precursor to tryptophan and many other products, and propose 4 designs here.

The workflow to optimize strain production while maintaining as close a metabolic state to a healthy non-production state seems like quite a nice concept. The authors generated an ensemble of kinetic models to account for parameter uncertainty and several alternate designs that seem promising. While I am unsure why Network Response Analysis is a core part of the workflow (see comment below), the remainder of the steps are a logical set of kinetic model generation and testing steps to evaluate possible designs.

Drawbacks of the work seem to be that 1) the workflow is heavily reliant on previously developed components and as such the novelty of the workflow itself is unclear, and 2) there is no experimental validation of solutions beyond consistency with prior literature, and thus the accuracy of the method remains unclear, and 3) the rejection sampling utilized in model selection seems to be somewhat fragile and vulnerable to overfitting. If the authors could experimentally validate their novel designs with a systematic approach that demonstrates that their method outperforms other possible rational designs that fail for reasons predicted by the model, this would be an outstanding work. I understand that this is a large amount of work, but without more testing of predictions, it is difficult to assess whether this workflow and the models themselves are truly successful. As is, it is a nice application of kinetic modeling that feels as yet under-proven. See my specific comments below.

Major Comments

- The key theme of the design workflow is to increase productivity while maintaining as close a flux state to the reference as possible. I understand the rationale behind this – altering concentrations of metabolites that have a signaling or allosteric role could have unintended off-target effects on metabolism and growth. Still, it might be expected that to achieve high productivity, carbon must be significantly rerouting towards the compound of interest. How can the authors balance the desire to maintain a similar state with the need to shift the state around desired pathways? It seems like there would be a continuity of alternate solutions that could be designed, balancing between these two objectives. In the end is this balance not somewhat arbitrary? Are there clear alternatives that are high and low similarity to the reference state, which have equivalent effect on productivity? The authors mention some such examples but more exploration would be warranted as this is a key concept in the work. I end up somewhat unclear on how useful this metric is in practice in selecting designs – maybe a sensitivity analysis would be helpful in this regard as well
- A common issue with this type of work that is difficult to overcome in practice - validation of their approach feels somewhat thin. The authors claim that ‘devised strategies include 8 previously validated targets’, but not sure if this counts as an ‘enrichment’ given a limited reasonable pool, heavy exploration of this pathway previously by metabolic engineering groups, and the potentially infinite space of solutions explorable by computational models. If the authors could more quantitatively demonstrate that their designs agree with previous experimental data, this would go a long way.
- Figure 5 represents some valuable experimental validation, but if Panel A and B are intended to be compared, the y axes should of course have the same scaling – demonstrating qualitative agreement is fine but should not be oversold. This type of analysis would also benefit from additional exploration of alternate designs to somehow determine whether the model is specific in its predictions.
- The rejection sampling used by the authors would seem to benefit from further exploration. If I understand correctly, the authors selected their set of kinetic models by identifying the subset of models that exhibited behavior that matched measured external concentrations over time. Is there any independent validation analyses that could be done here that would suggest that the selected models are not overfit?
- Related to the rejection sampling, I am curious about the relatively arbitrary number of models explored: 800000 models initially sampled, down to 212, down to 10. Presumably the final number of models is insufficient to explore the parameter space of a model containing over 120 reactions. What are the limitations here in exploring the parameter space more exhaustively? Is it time-consuming to

generate these models? Do the authors have difficulty generating models with meaningfully different parameters, i.e. does the explored parameter space converge?

- Is the rejection sampling procedure vulnerable to finding no models that satisfy the criteria? Are there any ways to guarantee that a model could be found, or enforce certain criteria a priori to increase the 'hit' rate of the sampling procedure? Starting with 800000 models and ending up with 10 seems somewhat discouraging.

- I am struggling to understand the role that Network Response Analysis plays in this workflow. With an ensemble of kinetic models, it would be straightforward to directly simulate enzyme perturbations without need for linear control coefficients fit within a constraint-based optimization. The authors suggest that such kinetic simulations would be computationally taxing, but could they be more quantitative about this limitation? The final designs from NRA seem to be relatively few interventions anyways, so it is not clear to me that for example, sampling from a pool of 3 enzyme perturbations would be truly too much for a modern server to handle. Additionally, presumably enzymes unrelated to the target pathway could be excluded from this exploration, and heuristic searches could be employed as well to identify effective kinetic perturbations.

- Related to the above point, it would be helpful to see a more explicit formulation of the NRA optimization in the methods, including how control coefficients are used. I read the original (recent) paper but a large number of alternative objectives were presented there and I left still unclear on how exactly it works. Could the authors please include the full MILP problem formulation in the Methods? For example, I do not understand how an MILP can enumerate all possible quantitative enzyme interventions. I can imagine how it can select enzymes to over or under express as binary variables, but the amount that you over or under express them could vary infinitely (within physiological ranges) – how would an MILP explore this space? Maybe I missed where this is explained.

Minor Comments

- The supplement containing information on kinetic mechanisms and regulation of enzymes would be much improved with citations.

- Maybe I missed it in the methods, but the authors simulate a bioprocess over 60 hours – how do they kinetically model growth and increased biomass in the culture? There is biomass over time data in Figure 2B, but it is unclear to me how their model predicts it correctly. Does their model truly identify growth limiting metabolites over time? This alone seems like it would be a remarkable achievement.

- For the filtering criteria, I did not understand why it is important that the dominant time constant $> 5x$ faster than the doubling time of the cell. What is this basis for this? If I understand it, this step eliminated 90% of models, which seems quite impactful. What are the slower dynamics in these? Would they actually be problematic or are they numerical artifacts around peripheral pathways? Does including these models actually affect anything? It would be nice to see this explained in more detail.

- What was the logic behind limiting enzyme perturbations to 5-fold? Is this the maximum observed among previous experimental interventions? Typical bacterial transcriptional regulation extends well beyond this.

Reviewer #3:

Remarks to the Author:

Narayanan et al present NOMAD, a workflow that take advantage of the large set of computational tools and methods developed by Hatzimanikatis groups in the last years for the construction and applications of large scale dynamic metabolic models. NOMAD is developed as an interesting complement of NRA facilitating the screening of optimal overproducing designs by scout the solution space. Main innovation of NOMAD relies in the inclusion of the concept of minimal phenotype disruptions similar to MOMA did with stoichiometric models. Additionally, the workflow allows the assignment of the robustness and sensitivity of the predicted design. The work is timely and meritorious since it includes the important concept of minimal phenotype disruption in strain designing. The manuscript is well structured and written. However, I find the work is somewhat

incomplete since it has been poorly validated with a single case of study and the quantitative predictions shown a limited accuracy. A more complete experimental validation of some of NOMAD-based predictions would be needed to significantly increase the impact of NOMAD as a broad and widely method in metabolic engineering.

Major comments

1. Pg. 13-14, "NOMAD performs well compared to experimental strategies for anthranilate production"

The authors perform here an interesting validation of NOMAD approach by constructing mutant in silico models base on experimental data. However, and despite NOMAD qualitatively captures in vivo results, their quantitative predictions are far away from the experimental data. For instance, the models largely fail into predict the impact of the double *aroG/tktA* mutation and while only a slight increase was predicted the in vivo data showed a 2.5-fold increase. The original in vivo study from Balderas-Hernandez shown additionally data of biomass production and substrate consumption. The comparison performed by the authors would be greatly enriched if they show the prediction of biomass production and glucose consumption of their strains in silico.

By the way, since NOMAD aim minimal impact on phenotype would be interesting to see whether the higher production of anthranilate is due growth performance maintenance.

2. Figure 1 shows NOMAD workflow, however I find the figure somewhat poor and not very descriptive. The figure could be greatly improved by including the different methods/tools in charge of each step as they are described in methods. Also the inclusion in the figure of the available software will facilitate a better understanding of NOMAD's novelty.

3. I found interesting the analysis of robustness and sensitivity analysis of the top 5 strategies which largely could facilitate the implementation of those more suitable designs. However, the real value of any computational metabolic engineering method is its effective validation under experimental conditions. As indicated above, NOMAD strongly underestimated the in vivo production of antranilate by the double *aroG/tktA* KO. Therefore, can the authors be sure that NOMAD is not underestimating the production of the designs shown in figure 4?

By the way, the upregulation of DDPa together the downregulation of GLUDy seems a very robust designs. What is the production of this double mutant when compared with the top 5 designs which include an additional reaction target?

4. Since the number of targets addressed in the NRA designs is reasonable (3), and multiple and efficient genetic tools are available for *E. coli*, the implementation of some the designs from the different groups identified in figure 4 will be very interesting in order to validate the whole NOMAD approach in term of production and phenotyping maintenance.

Authors' response to the reviewers' comments

We thank the reviewers for their insightful and constructive comments that have helped us improve the manuscript's quality and clarity. To address reviewers' remarks, we have performed several additional studies, modified the main text, added several Supplementary Notes (VI – X), and elaborated on unclear parts. Our detailed point-by-point responses to the reviewers' comments are below, highlighted in blue. We have numbered reviewers' comments to facilitate communication and referring.

In addition, the revised manuscript is available in two versions: a clean copy and a version with all alterations highlighted in blue.

REVIEWER COMMENTS

Reviewer #1 (Remarks to the Author):

In this work by Narayanan and coworkers they report a new computational framework for strain design called NOMAD. This method makes use of the advantages of kinetic models for metabolic engineering, such as capturing dynamic properties of the system and considering enzyme and metabolite concentration changes, however it reduces the search space to simplify the amount of possible solutions. The framework is novel and provides the basis for a design principle when using kinetic models: to seek the minimal deviation from the wild type phenotype. The presented data support the conclusions. They demonstrate how this design principle holds, then I consider that this could be a resource with great value to the design community and I can see how it will speed the DBTL cycle.

Major comments:

1. The main downside of this work is that it still requires enormous effort from the user to be implemented, such as but not limited to: it make use of previous methods (such as many kinetic model builds, NRA, MOMA etc.), It relies on the generation of thousands of kinetic models, great amounts of physiological and omic data, manual (or expert driven) quality checks and so on. Then it generates many possible solutions, and it is difficult to prioritize them. It seems that it will still be very difficult to be adopted by the average user.

Although kinetic models provide many benefits for comprehending and designing cellular processes, constructing them is a complex undertaking due to their coupling of several layers of biological information into a single mathematical framework. As a result, all existing kinetic modeling techniques in the field are intricate (e.g., Hu&Dinh et al., *Met Eng*, 2023, and Martin et al., *Met Eng*, 2023).

It should be noted that flux balance analysis (FBA) was not a widely used research method in this field 15-20 years ago, and it required considerable expertise and great amounts of data at that time. Today, it has become a common approach and is taught at the bachelor's and master's levels in many universities. The field of kinetic modeling is constantly evolving, and as more groups gain the required knowledge, these techniques are becoming more widespread. With the rise of a new generation of interdisciplinary researchers, it is anticipated that these methods will become standard practice.

In response to the reviewer's comment on the need to prioritize multiple solutions, many current kinetic modeling approaches create a population of models to account for uncertainty resulting from a lack of experimental observations and parameters, and we undeniably need to rank these models. Our proposed framework is fully compatible with these methods; it effectively prioritizes the resulting solutions and allows for the utilization of various ranking criteria (please also refer to our

response to reviewer remarks 3 and 4). To clarify this, we have modified the "Robust and implementation-suitable designs" subsection in the revised manuscript. Finally, our open-source Python code, which contains a detailed workflow with thoroughly documented strain design scripts, ensures that readers can effortlessly employ this method. Moreover, all the code for the methods used by the workflow, i.e., SKiMPy and pyTFA, is open source.

2. In this case they begin with a large population of kinetic models "with experimentally observed omics and cultivation data, physicochemical laws, network topology, and regulatory interactions." What is the perspective to use this method in non typical growth conditions or non-model organisms?

NOMAD can be employed to examine non-model organisms even with limited data available. The method is conceived to investigate a wide range of physiological conditions and organisms. It is capable of integrating various types of data, including growth rates and/or uptake/secretion rates. If certain data is not accessible, NOMAD permits sampling missing quantities within reasonable limits, such as the minimum and maximum metabolite concentrations observed in the organism under study or in the same genus. Additionally, NOMAD can be applied to any stoichiometry as long as the model is elementally and carbon balanced. No specific "minimum" amount of data is required to utilize NOMAD, and the more data accessible, the more precise the solution space will be.

3. The experimental validation in this case is fortunate in the shikimate pathway has been extensively explored for the production of several compounds so there are many references to show how the many combination of enzyme down and upregulation has been or not successful, however an average user of this method should prioritize among possible designs with out such information? We appreciate that the reviewer raised this question as it aligns with the objective of our work, which is to determine how kinetic models can produce meaningful and practical design options without experimental information on successful design choices.

Throughout our studies, we exclusively incorporated available reference or wild-type strain data. Information about previous successful experimental designs was used *only* to compare with NOMAD-devised designs and validate them.

Although it was not explicitly stated in the manuscript, the output of our pipeline is a ranked list of designs based on two criteria: (i) performance achieved in the NRA optimization and (ii) performance and robustness in bioreactor simulations.

Since bioreactor simulations closely simulate real-world conditions, we utilize the second criterion for the final ranking while rejecting poor performers from the first. This way, the millions of potential enzymatic intervention combinations are reduced to just a few viable options.

To clarify the above-stated points, we have modified the subsection Robust and implementation-suitable designs in the revised manuscript and reorganized the subsection Robust strain design using kinetic models by introducing the subsubsection Two-stage ranking of designs.

4. Can the authors provide another use case? and would it be desirable to provide a straightforward method to rank the provided designs so the experimental validation could start with the best cases? We are not entirely sure we have comprehended the reviewer's comment on "another use case." Nevertheless, if our understanding is correct and the reviewer asks for another study, we do not have one. However, after posting our preprint on bioRxiv.org, we were contacted by experimental groups that have expressed interest in using our framework in their experimental systems including *E. coli*, *S. cerevisiae*, and an autotrophic organism.

NOMAD aims to provide a set of ranked designs with a high likelihood of success in experimental implementation. However, we realized that our manuscript did not effectively convey this idea. During the screening, each design undergoes a sequence of tests, with yield being evaluated first

while also adhering to constraints on minimal perturbation of phenotype. The designs are then ranked, and only the top-ranked designs meeting a minimum yield requirement proceed. Similarly, designs are ranked in the context of bioreactor simulations. As stated in our response to the reviewers' remark 3, we have revised the manuscript to clarify this point.

While NOMAD ranks the designs based on a set of quantitative criteria, it is usually the case that the people who work on the problem and have expertise in the biological aspects of the problem might reprioritize among the top solutions based on expert knowledge, insight, and experience which cannot be immediately captured in the form of a mathematical model. However, the value of our method is that it offers a limited number of suggestions that satisfy constraints that are challenging to account for by conceptual design alone.

It's worth mentioning that the proposed method is highly flexible, as it allows for a wide range of design ranking objectives. For instance, users can select yield, specific productivity, uptakes, or a combination of goals as their design objective. The suggested ranking is just one of many reasonable objectives.

Finally, to our knowledge, this work is the first successful instance where a mechanistic model of such complexity has accurately reproduced bioreactor fermentations while remaining in line with the available data on metabolomics, fluxomics, thermodynamics, and physiology.

Minor:

5. Abstract: where the six orders of magnitude come from?

Compared to an exhaustive enumeration of designs, the six orders of magnitude reduction in the computational cost stems from our use of the NRA optimization to trim down the number of designs to perform nonlinear simulations. In the introduction, we provided a back-of-the-envelope calculation to demonstrate this:

"For example, to explore all possible strategies for manipulating (increasing or decreasing) the activities of five enzymes within a middle-sized metabolic network of 200 reactions (catalyzed by 200 enzymes), exhaustive enumeration would require performing more than $8.3 \cdot 10^{10}$ simulations."

Utilizing optimization techniques within the design space, our proposed framework can reduce this number to less than 1000 simulations, thus reducing the computational cost by more than six orders of magnitude. The amount of time saved varies depending on the specific study, but it can be substantial.

6. Methods: Briefly define why it is important to retain only the models whose jacobian matrix had all negative eigenvalues

All putative kinetic models created in this study capture the experimentally observed quasi-steady-state of metabolic fluxes during growth. The observed quasi-steady-state of metabolic fluxes implies the system must be "locally stable", i.e., after a slight perturbation of concentrations or metabolic rates they must return to initial steady state. Therefore, verifying how these models behave when exposed to slight perturbations around the observed steady state is important. Indeed, if a model that has been perturbed does not return to the observed steady state, it is highly likely that the model is not accurately describing the experimentally observed steady-state behavior.

Through the use of linear algebra and the theory of dynamical systems, we know that the eigenvalues of the Jacobian matrix can provide information on the local stability of the system around the steady state. In particular, if all eigenvalues have negative real parts, the steady state is locally stable, and nearby trajectories will converge towards it over time. Conversely, if at least one eigenvalue has a positive real part, nearby trajectories will diverge from the steady state over time. As a result, in our study, we have only kept models whose Jacobian matrix had all negative eigenvalues.

As suggested by the reviewer, we have modified the Screening kinetic models subsection of the Methods and added the preceding explanation.

7. Methods: the steps to screen the kinetic models are not clear, it seems that they only describe the linearized dynamics

As suggested by the reviewer, we have described the screening procedure in more detail under Methods, Screening kinetic models.

Reviewer #2 (Remarks to the Author):

In this work, the authors develop a workflow called NOMAD (NONlinear dynamic Model Assisted rational metabolic Engineering) to explore the design space within a kinetic model of metabolism. A major focus of this workflow is maintaining the physiological state near a healthy one (growth rate), practically done by maintaining fluxes and metabolite concentrations near the reference state. They perform this analysis for the production of anthranilate, a precursor to tryptophan and many other products, and propose 4 designs here.

The workflow to optimize strain production while maintaining as close a metabolic state to a healthy non-production state seems like quite a nice concept. The authors generated an ensemble of kinetic models to account for parameter uncertainty and several alternate designs that seem promising. While I am unsure why Network Response Analysis is a core part of the workflow (see comment below), the remainder of the steps are a logical set of kinetic model generation and testing steps to evaluate possible designs.

Drawbacks of the work seem to be that 1) the workflow is heavily reliant on previously developed components and as such the novelty of the workflow itself is unclear, and 2) there is no experimental validation of solutions beyond consistency with prior literature, and thus the accuracy of the method remains unclear, and 3) the rejection sampling utilized in model selection seems to be somewhat fragile and vulnerable to overfitting. If the authors could experimentally validate their novel designs with a systematic approach that demonstrates that their method outperforms other possible rational designs that fail for reasons predicted by the model, this would be an outstanding work. I understand that this is a large amount of work, but without more testing of predictions, it is difficult to assess whether this workflow and the models themselves are truly successful. As is, it is a nice application of kinetic modeling that feels as yet under-proven. See my specific comments below.

Detailed point-by-point responses to the reviewer's comments are presented below.

Major Comments

1. The key theme of the design workflow is to increase productivity while maintaining as close a flux state to the reference as possible. I understand the rationale behind this – altering concentrations of metabolites that have a signaling or allosteric role could have unintended off-target effects on metabolism and growth. Still, it might be expected that to achieve high productivity, carbon must be significantly rerouting towards the compound of interest.

We agree with the reviewer's statement that directing carbon toward the target compound is critical for achieving high productivity, and NOMAD was specifically designed with this factor in mind. The proposed method offers the capability to apply design constraints on *both* metabolite concentrations and metabolic fluxes, which distinguishes it significantly from the MOMA method, which can only limit a flux state. Additionally, we can decide which fluxes and concentrations to restrict and to what degree. For example, one could leave fluxes and concentrations in the production pathway unconstrained and constrain the other parts of the metabolic network. In the provided study, we have enforced constraints to retain metabolite concentrations reasonably close to the reference state while leaving fluxes unconstrained, enabling the redirection of carbon toward the production of anthranilate.

The method also identifies manipulations that will increase the overall throughput flux that will be “rerouted” to the product without compromising other downstream fluxes. Changing upstream fluxes could also increase the accumulation of metabolites. Therefore, the method, which also constrains metabolite concentrations, identifies the right combination of changes that can increase throughput flux without an unphysiological change in metabolite concentrations.

How can the authors balance the desire to maintain a similar state with the need to shift the state around desired pathways? It seems like there would be a continuity of alternate solutions that could be designed, balancing between these two objectives. In the end is this balance not somewhat arbitrary?

The proposed method excels in precisely this aspect. Its formulation allows the integration of various cellular requirements defined by the user while identifying conflicting requirements and facilitating trade-offs between them. As a reasonable cellular requirement, we sought to obtain the highest possible productivity without compromising the robustness of the engineered strain. As we have demonstrated in our studies, we can indeed have a higher specific productivity of anthranilate if we compromise growth. However, the final anthranilate titer will be low if we compromise growth too much. Importantly, the computational efficiency of the proposed framework allows for exploring alternative design choices in a ranked order and converging toward the most optimal one.

Are there clear alternatives that are high and low similarity to the reference state, which have equivalent effect on productivity? The authors mention some such examples but more exploration would be warranted as this is a key concept in the work. I end up somewhat unclear on how useful this metric is in practice in selecting designs – maybe a sensitivity analysis would be helpful in this regard as well

Following the reviewer's recommendation, we studied how productivity changes depending on two key factors: the degree of similarity to the reference state and the strength of the applied engineering intervention. Essentially, we repeated the strain design process for 48 different combinations of permissible fold changes in concentrations (2, 3, 4, 5, 6, 7, 8, 9, 10, 12, 15, and 20) and enzyme activities (2, 3, 5, and 10), and generated the top-performing design for each model. We then applied these top-performing designs to their respective models and simulated their behavior in batch fermentation. This allowed us to assess the sensitivity of (i) log-linear NRA design predictions and (ii) actual nonlinear performance to the degree of similarity (high and low) to the reference state.

As expected, our log-linear NRA predictions showed that increasing the deviation of engineered strains from the reference phenotype resulted in a larger improvement in anthranilate production.

In nonlinear batch reactor simulations, however, the extent to which we can deviate from the reference phenotype without compromising productivity depends on the strength of the enzymatic interventions. Maintaining the engineered strains close to the reference phenotype generally results in better anthranilate titers, particularly when more substantial enzymatic interventions (10 or 20-fold changes in enzyme activity) are applied. At these over/under-expression levels, deviating too far from the reference strain results in a significant decrease (up to 50%) in production rate. On the other hand, if we apply less intense interventions (with a maximum of 2-fold and 3-fold enzyme activity changes), we observe a less significant reduction in production rate (about 10%) when the strains are permitted to deviate further from the reference phenotype. Additional information on this study can be found in Supplementary Note VI.

2. A common issue with this type of work that is difficult to overcome in practice - validation of their approach feels somewhat thin. The authors claim that ‘devised strategies include 8 previously validated targets’, but not sure if this counts as an ‘enrichment’ given a limited reasonable pool, heavy exploration of this pathway previously by metabolic engineering groups, and the potentially infinite space of solutions explorable by computational models. If the authors could more quantitatively demonstrate that their designs agree with previous experimental data, this would go a long way.

We want to emphasize that in the two studies conducted for experimental validation, i.e., comparison against experimentally implemented targets (i) around shikimate metabolism and (ii) anthranilate production, we *only* integrated limited data (the cultivation data) on the wild-type strain – and yet, we have obtained convincing results.

Indeed, in the study (i), without using any prior knowledge of previously experimentally tested targets such as DDPA regulation, NOMAD captured 8 such targets. It's worth noting that among millions of possible combinations, these 8 targets were included in the highest-ranked designs. Moreover, NOMAD designs captured not only the experimentally tested targets of genetic manipulation but also the direction of these manipulations (whether it was over or under-expression).

In the study (ii), we accurately captured the relative performance enhancements of two experimental strains but not their exact values. Nonetheless, these results suggest that designs created using NOMAD would improve cellular performances in practical experiments. Please refer to our reply to Reviewer 3, remark 1, for additional information.

In general, given the limited experimental data and the uncertainty, we should not expect to have the exact ranking right, but rather the ID and the number of the top targets that would lead to desired cellular performances.

The additional study conducted for this revision reveals that by increasing the maximal velocities of pathway enzymes (SHKK, DHQS, ANS) and decreasing the activity of GLUDy to eliminate rate-limiting steps in the shikimate pathway, our models could quantitatively capture the experimental performance improvements achieved by the strain *trpD9923 aroG^{fbr} tktA* (Supplementary note VIII). This illustrates how additional expert or experimental evidence can be used to improve the reliability and accuracy of predictions.

Considering the reviewer's remark about 'enrichment', NOMAD is capable of capturing network-wide effects of genetic manipulation; consequently, it allows us to discover novel but also non-obvious targets that can be neglected by the experts. It is also proficient at detecting *combinations* of enzymes that synergistically affect the desired performance. In other words, it can also detect situations where enhancing the production of a target compound by modifying specific enzymes may have adverse effects on other essential properties, such as carbon uptake. It also should be noted that conceptual designs, usually used in experiments, mainly consider stoichiometry and maybe some conceptual MCA analysis. However, such manual, heuristics-based conceptual design cannot integrate the full kinetic coupling *and* the constraints in concentrations and fluxes considered by our method.

As per the reviewer's recommendation, we compared our designs against the experimentally tested designs in more detail. While we could qualitatively compare the modifications suggested by NOMAD with the experimental ones, we were unable to perform a quantitative analysis since the extent of up/down-regulation implemented in the experiments was not reported. Nevertheless, we have slightly revised the text in the subsection "NOMAD qualitatively captures experimental strategies for anthranilate production".

It should be emphasized that bioreactor simulations and predictions utilizing nonlinear kinetic models of metabolic networks coupled to growth in bioreactor have not been previously attempted. Rather than employing phenomenological formulas like Monod growth kinetics, the growth process is represented by a biomass equation that models cellular biosynthetic requirements, and the obtained growth curves emerge from the underlying metabolic processes.

3. Figure 5 represents some valuable experimental validation, but if Panel A and B are intended to be compared, the y axes should of course have the same scaling – demonstrating qualitative agreement is fine but should not be oversold. This type of analysis would also benefit from additional exploration of alternate designs to somehow determine whether the model is specific in its predictions.

We fully concur with the reviewer's suggestion that the scaling of y-axes should be identical. The lack of uniformity was simply an oversight and not a deliberate attempt to exaggerate the effectiveness

of our approach. In fact, we explicitly stated in the manuscript that our models could qualitatively capture the performance of the experimental strains but not quantitatively. In the revised manuscript, y-axes in Figure 5 have the same scaling.

In addition, we conducted a study to examine why our models were unsuccessful in reproducing the experimentally observed titers for the second strain, which targets transketolase. Our study indicates that the models have slightly underestimated the maximal velocities of the enzymes involved in the shikimate pathway (see also our response to your remark 2). We were able to replicate the same titers (0.7 g/L) as those observed experimentally for the double mutant by increasing the V_{max} values of these enzymes. We provided the details of this study in supplementary note VIII.

4. The rejection sampling used by the authors would seem to benefit from further exploration. If I understand correctly, the authors selected their set of kinetic models by identifying the subset of models that exhibited behavior that matched measured external concentrations over time. Is there any independent validation analyses that could be done here that would suggest that the selected models are not overfit?

The reviewer is correct in stating that the kinetic models selected for our design demonstrate behavior that aligns with the temporal evolution of measured extracellular concentrations. The concerns raised by the reviewer regarding overfitting are important when striving to ensure the models agree with experimental data. However, it is worth noting that this method differs from traditional parameter estimation methods, where the parameters are determined by minimizing a measure of how well a model fits growth and uptake curves. In our case, parameters are unbiasedly sampled and then selected based on several criteria.

We have recognized the issue of overfitting an underdetermined problem, which is why we also sample a population of parameter sets (models) consistent with the observed solutions. By increasing the number of models/parameter sets, which is also a feasible aspect of our method, we could further address the overfitting issue.

Moreover, we reduce the likelihood of overfitting in our models by enforcing consistency across multiple criteria. These criteria ensure that the model is consistent with (a) available steady-state cultivation data during exponential growth; (b) experimentally measured intracellular metabolomics and fluxomics data; (c) metabolite concentration and metabolic fluxes that are thermodynamically feasible, as well as related physicochemical properties such as energy charge and redox ratio; (d) local stability around the steady-state; (e) dynamic properties of cellular processes; (f) robustness to perturbations in metabolite concentrations and enzyme activities; and (g) temporal changes in glucose uptake, anthranilate production, and growth. This approach also guarantees that our models more accurately depict cellular metabolism, which is one of the prominent features of our methodology.

In this case, we could implement the experimental targets specified in the paper by Balderas-Hernandez et al. Our models were able to replicate the trends reported experimentally *without being specifically tailored to fit those results*. This suggests our models are not overfitting the reference physiology they were calibrated on.

5. Related to the rejection sampling, I am curious about the relatively arbitrary number of models explored: 800000 models initially sampled, down to 212, down to 10. Presumably the final number of models is insufficient to explore the parameter space of a model containing over 120 reactions. What are the limitations here in exploring the parameter space more exhaustively? Is it time-consuming to generate these models? Do the authors have difficulty generating models with meaningfully different parameters, i.e. does the explored parameter space converge? Is the rejection sampling procedure vulnerable to finding no models that satisfy the criteria? Are there any ways to guarantee that a model could be found, or enforce certain criteria a priori to

increase the 'hit' rate of the sampling procedure? Starting with 800000 models and ending up with 10 seems somewhat discouraging.

In our response to the reviewer's remark 4, we explained that models capable of accurately predicting cellular responses to genetic modifications must meet various criteria. Due to the underdetermined nature of the system and considerable uncertainty caused by a lack of data and a large model size, countless models meet the imposed criteria. To find such models, we initially start with a sizable population of dynamic models and then trim it down, as stated in the manuscript. However, as the reviewer noted, this task is time-consuming with current kinetic modeling approaches. For example, Gopalakrishnan et al. (*Met Eng*, 2020) reported that they could parameterize 100 models in 48 hours, having sped up their parameterization by three orders of magnitude. Although the ORACLE framework used in our work is more efficient and generates diverse, meaningful parameters, creating and pruning models is still costly because we are exploring the space of 613 kinetic parameters (parameterizing 123 reactions and 90 mass balances). Similar to the reviewer, we have also asked ourselves how we can improve the efficiency of generating "good" models. To address this issue, we have developed several methods, such as iSCHRUNK (Andreozzi et al., *Met Eng*, 2016; Miskovic et al., *PLoS Comp Bio*, 2019), REKINDLE (Choudhury et al., *Nat Mach Intel*, 2022), and RENAISSANCE (Choudhury et al., [biorxiv.org](https://doi.org/10.1101/2023.03.15.531111), 2023), which utilize machine learning. The latter two methods are particularly efficient in generating "good" models by performing stratified sampling in kinetic space areas that meet the abovementioned criteria.

6. I am struggling to understand the role that Network Response Analysis plays in this workflow. With an ensemble of kinetic models, it would be straightforward to directly simulate enzyme perturbations without need for linear control coefficients fit within a constraint-based optimization. The authors suggest that such kinetic simulations would be computationally taxing, but could they be more quantitative about this limitation? The final designs from NRA seem to be relatively few interventions anyways, so it is not clear to me that for example, sampling from a pool of 3 enzyme perturbations would be truly too much for a modern server to handle. Additionally, presumably enzymes unrelated to the target pathway could be excluded from this exploration, and heuristic searches could be employed as well to identify effective kinetic perturbations.

A short answer about the role of NRA in the proposed framework is given in our response to reviewer 1, remark 5. In more detail, the studied model has 120 possible enzymatic targets resulting in 240 possible enzymatic interventions (each enzyme can either be down or upregulated). For targeting 3 enzymes simultaneously, we would need 2.27 million simulation runs to explore all possible combinations.

Furthermore, we must assess different levels of up- or downregulation for each combination of three enzymes since the relative amount of relative expression (up and down) is also very important to achieve the proper balancing of hundreds of metabolites and fluxes. Considering that we must also repeat simulations for each of the ten models, the required number of simulations rapidly increases to tens and even hundreds of millions. Given that each nonlinear simulation requires between 30 seconds to 1 minute to complete on a workstation, the execution of 100 million simulations would take at minimum 95 years without parallelization.

In contrast, computing all the flux and concentration control coefficients for each of the ten models takes less than a minute. Subsequently, the NRA method takes 3-5 minutes to enumerate all feasible design options, as it incorporates the design constraints and objectives within the strain design problem formulation. Following this, obtaining predictions for each design in each model takes less than an hour without parallelization. Since we conduct nonlinear simulations for only the top 10 (or 20 or 100) performing designs, even after conducting rigorous sensitivity analyses of the designs across all parameter ranges the entire strain design workflow can be completed within less than 48 hours without parallelization. The gain in computational efficiency becomes more pronounced with larger models.

Considering the comment about discarding enzymes unrelated to the pathway from the design considerations, one of the key advantages of using kinetic modeling approaches is the ability to identify seemingly unrelated targets that are crucial to attaining the design objective due to network interactions and regulation. For example, enzymes that might appear to be unrelated to the pathway of interest might be important for maintaining physiological levels of metabolites, energy, redox, and fluxes. From our previous experience, we suggest not excluding any enzymatic target beforehand from the design considerations. Additionally, even if the designer decides to ignore some metabolic subsystems to narrow the search space, performing brute-force simulations for larger-scale kinetic models can still be prohibitively expensive.

7. Related to the above point, it would be helpful to see a more explicit formulation of the NRA optimization in the methods, including how control coefficients are used. I read the original (recent) paper but a large number of alternative objectives were presented there and I left still unclear on how exactly it works. Could the authors please include the full MILP problem formulation in the Methods? For example, I do not understand how an MILP can enumerate all possible quantitative enzyme interventions. I can imagine how it can select enzymes to over or under express as binary variables, but the amount that you over or under express them could vary infinitely (within physiological ranges) – how would an MILP explore this space? Maybe I missed where this is explained.

We thank the reviewer for this suggestion. We added Supplementary Note X with an explicit formulation of the optimization used in this work.

It is crucial to note that in MILP problems, the objective function and the constraints, both equalities and inequalities, are linear. The presence of integer variables facilitates optimization across discrete scenarios, such as determining whether to up or downregulate a gene to achieve an objective.

In this context, multiplicity, in general, comes in two types: (i) in alternative integer solutions, and (ii) within each integer solution, there are alternative solutions in continuous variables. The branch and bound type of MILP solvers, e.g., in CPLEX we were using, first find the optimal solution by relaxing integer variables, then fix the optimal solution, and ask what is the minimum number of enzymes, which, if manipulated, can achieve this goal. Following this, we impose a new constraint that asks how many other sets with this minimal number can meet the design specifications. In general, for this type of problem, the number of alternatives is countable.

Minor Comments

8. The supplement containing information on kinetic mechanisms and regulation of enzymes would be much improved with citations.

We thank the author for pointing this out. In the ORACLE framework and its SKiMpy tool, we employ generalized approximations of enzymatic mechanisms, such as generalized reversible Hill (Hofmeyr and Cornish-Bowden, *Bioinformatics*, 1997) and convenience kinetics (Liebermeister and Klipp, *Theor Biol and Medical Model*, 2006). Additionally, the framework allows using other kinetics laws, including reversible and irreversible Michaelis-Menten kinetics, Uni-Bi, Bi-Uni, random Bi-Bi, ordered Bi-Bi, Bi-Ter, Ter-Bi, and more (I.H. Segel, book, 1975). As for the regulations, we have obtained all of them except for two from Khodayari and Maranas (*Nature Communications*, 2016), which has already been referenced in the main text. We added the regulation of DDPA by phenylalanine as *aroG* is the major isoform of the three *dahp* synthetase isoforms (Lin et al, *Inter. Journal of Bio Macromolecules*, 2012) referenced in the main text. We also added the regulation of ANS by tryptophan (Kwak et al, *Journal of Biochem and Mol Biology*, 1999) referenced in the text. Finally, the complete list of used kinetic mechanisms and enzyme regulations is provided in the supplementary material (reaction_information.xlsx).

9. Maybe I missed it in the methods, but the authors simulate a bioprocess over 60 hours – how do they kinetically model growth and increased biomass in the culture? There is biomass over time data

in Figure 2B, but it is unclear to me how their model predicts it correctly. Does their model truly identify growth limiting metabolites over time? This alone seems like it would be a remarkable achievement.

To represent biomass growth, we utilize a lumped biomass reaction that accounts for the biosynthetic requirements of the cell. We generate this reaction using lumpGEM, a custom tool developed in-house (Ataman & Hatzimanikatis, *PLoS Comp Bio*, 2017). The lumped biomass reaction can synthesize biomass building blocks using the core metabolites from our reduced model as precursors. We employ irreversible multi-substrate Michaelis-Menten to model the kinetics of this reaction. The reviewer's statement that the model captures growth-limiting phenomena over time is correct. As far as we know, this is the first instance where a mechanistic model of this degree of complexity has successfully replicated bioreactor fermentations while being consistent with the available metabolomic, fluxomic, thermodynamic, and physiology data.

10. For the filtering criteria, I did not understand why it is important that the dominant time constant $> 5x$ faster than the doubling time of the cell. What is this basis for this? If I understand it, this step eliminated 90% of models, which seems quite impactful. What are the slower dynamics in these? Would they actually be problematic or are they numerical artifacts around peripheral pathways? Does including these models actually affect anything? It would be nice to see this explained in more detail.

Metabolic processes occur on different time scales, ranging from milliseconds to minutes, and the specific time constants depend heavily on the organism. It is well-known that metabolic responses to perturbations settle down before cell division (Schaub & Reuss, *Biotechnol Prog*, 2008; Aboka et al., *FEMS Yeast Research*, 2009; Heinrich & Schuster, *The Regulation of Cellular Systems*, textbook).

According to dynamic systems theory, when a biological system exhibits biologically relevant aperiodic or damped oscillatory responses, it is expected to reach 99.3% of the steady-state value within 5 dominant time constants. Therefore, for a model to accurately depict biologically relevant processes, its dominant time constant should be at least 5 times faster than the cell's doubling time. Slow responses in peripheral pathways, particularly in biosynthetic pathways, can significantly impact the overall metabolic network, slowing down critical processes and ultimately resulting in a slower growth rate than experimentally observed.

As recommended by the reviewer, we have incorporated the discussed matter in the revised manuscript, the Methods section, subsection Screening kinetic models. More details on time constants are provided in Supplementary Note IX.

11. What was the logic behind limiting enzyme perturbations to 5-fold? Is this the maximum observed among previous experimental interventions? Typical bacterial transcriptional regulation extends well beyond this.

While some recent studies have reported significant increases in *in vivo* enzyme activity, ranging from 20-fold (Hou et al., *Protein Expr Purif*, 2013; Xiao et al., *Curr Opin Struct Biol* 2014) to 33-fold (Michener and Smolke, *Metab Eng*, 2012), following discussions with our experimental collaborators, we have reached the consensus that the existing technology could realistically achieve close to a 5-fold artificially induced increase in enzyme activity. That being said, if needed, users have the option to impose higher acceptable fold changes in enzyme activity.

Reviewer #3 (Remarks to the Author):

Narayanan et al present NOMAD, a workflow that take advantage of the large set of computational tools and methods developed by Hatzimanikatis groups in the last years for the construction and applications of large scale dynamic metabolic models. NOMAD is developed as an interesting complement of NRA facilitating the screening of optimal overproducing designs by scout the solution space. Main innovation of NOMAD relies in the inclusion of the concept of minimal phenotype disruptions similar to MOMA did with stoichiometric models. Additionally, the workflow allows the assignment of the robustness and sensitivity of the predicted design. The work is timely and

meritorious since it includes the important concept of minimal phenotype disruption in strain designing. The manuscript is well structured and written. However, I find the work is somewhat incomplete since it has been poorly validated with a single case of study and the quantitative predictions shown a limited accuracy. A more complete experimental validation of some of NOMAD-based predictions would be needed to significantly increase the impact of NOMAD as a broad and widely method in metabolic engineering.

Major comments

1. Pg. 13-14, "NOMAD performs well compared to experimental strategies for anthranilate production"

The authors perform here an interesting validation of NOMAD approach by constructing mutant in silico models base on experimental data. However, and despite NOMAD qualitatively captures in vivo results, their quantitative predictions are far away from the experimental data. For instance, the models largely fail into predict the impact of the double *aroG/tktA* mutation and while only a slight increase was predicted the in vivo data showed a 2.5-fold increase. The original in vivo study from Balderas-Hernandez shown additionally data of biomass production and substrate consumption. The comparison performed by the authors would be greatly enriched if they show the prediction of biomass production and glucose consumption of their strains in silico.

By the way, since NOMAD aim minimal impact on phenotype would be interesting to see whether the higher production of anthranilate is due growth performance maintenance.

Although our predictions of the double *aroG/tktA* mutation underestimate the experimentally achieved titer, as noted by the reviewer, it is crucial to acknowledge that the models correctly preserve all relative performance improvements. Indeed, the experimental results show that the *aroG/tktA* strain demonstrated a relative improvement of 141% compared to the wild-type strain, which is more than three times larger than the improvement observed in the *aroG* strain, which was 50.6%. Our predictions also show a similar pattern where the *aroG/tktA* strain had a relative gain approximately three times larger than the improvement observed in the *aroG* strain (14.3% vs. 4.57%). Based on these results, it can be inferred that the designs developed by NOMAD (Figure 5, blue) have the potential to improve the anthranilate titer even further when compared to the *aroG/tktA* strain.

It's also worth noting that our simulated *aroG/tktA* strains' growth of around 1.28g/L remained significantly closer to the reference strain's growth of 1.31 g/L (Supplementary Figure S.5) than what was observed in the experimental strain (0.95 g/L). As the reviewer suggested, maintaining growth closer to the reference strain in simulation than in the experiments could explain the observed quantitative discrepancy in the anthranilate titers. We provide the prediction of biomass production and glucose consumption of our strains in silico in Supplementary Note IV.

Moreover, motivated by the reviewer's remarks, we explored what additional transcriptional/enzyme activity changes could describe the experimentally observed increase in anthranilate for the double mutant targeting *aroG* and *tktA*. We used NRA to identify enzyme activity alterations, in addition to *aroG* and *tktA*, that would enable us to achieve the same anthranilate levels as the experimental strain. By increasing the activities of enzymes in the anthranilate production pathway (*SHKK*, *DHQS*, *ANS*) and decreasing the activity of *GLUDy*, we could obtain titers (0.7g/L) close to the experimental titers of 0.75 g/L. We also noticed that the final growth dropped to 1.05 g/L which is closer to the experimentally observed titer of around 0.93g/L. Details of the implementation and results have been added as Supplementary Note VIII.

Overall, considering the limited amount of data used to model the reference strain and the fact that we did not incorporate any data regarding the design strains, it is encouraging that kinetic models of this complexity could capture these trends in the first place.

2. Figure 1 shows NOMAD workflow, however I find the figure somewhat poor and not very

descriptive. The figure could be greatly improved by including the different methods/tools in charge of each step as they are described in methods. Also the inclusion in the figure of the available software will facilitate a better understanding of NOMAD's novelty.

We thank the reviewer for the suggestion, as it will enable readers to understand the proposed framework's conceptual flexibility better. We have modified the figure and its caption in the revised manuscript.

3. I found interesting the analysis of robustness and sensitivity analysis of the top 5 strategies which largely could facilitate the implementation of those more suitable designs. However, the real value of any computational metabolic engineering method is its effective validation under experimental conditions. As indicated above, NOMAD strongly underestimated the in vivo production of anthranilate by the double *aroG*/*tktA* KO. Therefore, can the authors be sure that NOMAD is not underestimating the production of the designs shown in figure 4?

Indeed, as mentioned in our response to the reviewer's remark 1, the kinetic models underestimate the impact of *aroG* and *aroG*/*tktA* intervention, and that the experimental titers of the NOMAD-devised strains will likely have superior performance to that suggested by the in silico simulations. However, given that NOMAD correctly predicts the relative changes in cellular responses, the ranking of the designs is expected to coincide with their experimental performances.

4. By the way, the upregulation of DDPa together the downregulation of GLUDy seems a very robust design. What is the production of this double mutant when compared with the top 5 designs which include an additional reaction target?

This is a great observation by the reviewer. Indeed, as the reviewer noticed, DDPa and GLUDy recurrently appear in many designs. As recommended by the reviewer, we incorporated the double mutant into all ten kinetic models and verified that it performs well compared to the triple mutant strains, with a median anthranilate titer of 0.41 (Supplementary Figure S.10) compared to titers of 0.4 – 0.41 for the triple mutants (Figure 7A). In terms of growth, the double mutant produced a median biomass titer of 1.278 g/L while the top 4 triple mutants had mean biomass titers of 1.26 – 1.29g/L. From the relatively similar performances of the double and triple mutants, it appears that DDPa and GLUDy carry the bulk of the work in terms of redirecting resources toward anthranilate production. The results of this study are incorporated in Supplementary Note VII.

5. Since the number of targets addressed in the NRA designs is reasonable (3), and multiple and efficient genetic tools are available for *E. coli*, the implementation of some the designs from the different groups identified in figure 4 will be very interesting in order to validate the whole NOMAD approach in term of production and phenotyping maintenance.

While we agree with the reviewer that further experimental validation of NOMAD designs would be highly valuable, it is important to note that such validation is beyond the scope of this manuscript. Along this line, following the publication of our preprint on bioRxiv.org, we are discussing with experimental teams who expressed a keen interest in applying our framework to their respective experimental setups, including organisms such as *E. coli*, *S. cerevisiae*, and an autotrophic organism.

Reviewers' Comments:

Reviewer #1:

Remarks to the Author:

I think this is a better version of the manuscript, most of my comments were addressed. However, all the reviewers brought the concern of more use cases to further validate the approach. This doesn't need to be experimental, they can find other published cases as they did with the anthranilate case. The authors argue that some experimental groups have already expressed interest in using their framework. I still think that this work would be much better presented if it presents more use cases.

Reviewer #2:

Remarks to the Author:

The authors have addressed my comments largely through text responses as opposed to additional analyses, but their sensitivity analysis on the degree to which the solutions can deviate from the reference point is appreciated. It does raise an additional question though - they find that larger perturbations result in poorer designs if I understand correctly, but are they sure that this isn't simply a result of relying on log-linear kinetics that are increasingly inaccurate for larger perturbations, or dependent on their predictions of growth rate based on the lumped reaction, which could have similar issues for larger perturbations? i.e. Is this not a problem with the limitations of their workflow more so than with large perturbations in general? The details of the designs were not discussed so it's difficult for me to determine the biochemistry of what is problematic in the large perturbation designs.

The remainder of their responses are logical but still feel overconfident given the thin validation. The workflow is still interesting and the idea of minimizing metabolic adjustment during designs is logical, but without further validation and given the continued numerical limitations (very few models evaluated in the end to assess alternate designs, dependence on linear control coefficients), it still feels unproven.

I think this work will still attract a lot of interest in the community because of the scale of modeling that is now possible - However, on the other hand I hope that kinetic modelers begin to take validation seriously because these models continue to grow in complexity and scale, making evaluation of the multitude of workflow decisions near impossible.

Reviewer #3:

Remarks to the Author:

I greatly appreciate the effort made by the authors to respond to my previous comments, but I feel that my main concerns related to this article still remain. Despite the complexity and complete computational work of the study, the final results, in terms of quantitative accuracy and validation, are, in my opinion, insufficient to recommend its publication in the current form.

Regarding the lack of quantitative accuracy, I agree with the authors that MONAD captures qualitative trends, but for high-quality kinetic models it would be expected more accurate quantitative predictions. In this sense, I appreciate the authors' additional analysis presented in Supplementary Note VIII, which identifies the source of discrepancies between model predictions and experimental data. However, rather than alleviating my concerns, this analysis has somewhat heightened them. The need for additional reactions to be overexpressed to fit the experimental data raises questions about whether the models used are overly constrained. Could this reduced degree of freedom in the models hinder the identification of better designs?

In terms of validation, I still believe that a proper experimental validation of NOMAD designs would be

desirable (only 3 genetic interventions are needed). While I understand that experimental validation may not be the primary focus of this work, I agree with reviewer 1 that addressing, at least one more case study, would enhance the scope and value of the proposed workflow.

Agree with the authors that rational strain design using kinetic models is really a holy grail in metabolic engineering. Therefore, I commend the authors for the tremendous effort done in this sense. However, I firmly believe that a more robust validation, either experimental or computational, is necessary to strengthen the findings.

Authors' response to the second round of reviewers' comments

Response to all reviewers:

The reviewers requested an additional validation study to enhance the value of the proposed workflow. This study needed to rely on pre-existing experimental research because this is meant to be a computational methods paper, and there was no experimental group part of this paper. We performed an exhaustive search of the literature to find experimental studies with data collected on the *E. coli* strains satisfying the following criteria:

- (i) The experimental study should provide batch fermentation data encompassing subsystems and production pathways contained within our models' stoichiometry (glycolysis, pentose phosphate pathway (PPP), the Krebs cycle, anaplerotic reactions, the shikimate pathway, and glutamine synthesis).
- (ii) The study should involve at least two strains as we calibrate the models using the reference strain and validate them by replicating the behavior of the recombinant strain.
- (iii) The study should provide fermentation curves for all involved strains, enabling us to validate our findings against empirical data.
- (iv) The study should have been conducted on a minimal media because our models do not involve uptake pathways for complex media, such as the uptake of amino acids when using yeast extract as a medium.

We could find 21 studies¹⁻²¹ that met (i) and (ii), with several sourced from a review²² of existing literature for the overproduction of shikimate-derived compounds. However, none of the analyzed studies met the abovementioned requirements. Most studies did not meet criterion (iv) as they utilized complex media containing yeast extract^{1,3-11,15,14,16-19,21}. Some studies also used M9 medium for certain cultures; however, they do not meet criterion (iv) since they only provided fermentation curves for the cultures using complex media.^{12,13,20}

Given the above, we address the reviewers' request for an additional validation case study that will demonstrate quantitative accuracy by returning to the work by Balderas-Hernandez et al. In the original submission and first revision, we built models (referred to as K_trpD9923 in the revised manuscript) calibrated on the reference strain only. In this revision, we constructed an enhanced set of models calibrated using data from: A – reference (W3110 trpD9923) and B - recombinant (W3110 trpD9923/pJLaroGfbr) strains. These models successfully reproduced the behavior of the third strain, C (W3110 trpD9923/pJLaroG^{fbr}tktA), thereby validating the model generation workflow (Figure 6 of the revised manuscript).

Simultaneously, we also directly addressed the reviewers' request to demonstrate not only the qualitative but also the quantitative accuracy of our models. The refined models accurately captured the anthranilate production in all 3 experimental strains. This study demonstrates that the quality of the models significantly improves with the integration of data from multiple physiologies. It also underscores the importance of extensive datasets for developing highly accurate metabolic models. Even for sophisticated modeling techniques, such datasets are essential for unraveling the complex, nonlinear nature of metabolic responses.

Interestingly, the NOMAD designs based on the models from the first study (K_trpD9923) performed well when applied to the enhanced models (eK_trpD9923), suggesting that the NOMAD-devised targets are robust across different sets of models. Finally, we also used eK_trpD9923 to suggest designs to improve anthranilate production in the double mutant, W3110 trpD9923/pJLaroG^{fbr}tktA.

We have substantially revised the Results section to incorporate the findings from our newly added case study. Specifically, we relocated the subsection 'Alternative routes for producing anthranilate' to Supplementary Note II. Additionally, we moved Figures 6 and 7 and their accompanying text to

Supplementary Note III. In their place, we introduced a new Results subsection titled 'Refined designs from models calibrated on reference and recombinant strains' physiology,' which includes two new figures. The new Supplementary Notes IX and X contain additional information on the clustering and prioritization of designs using these enhanced models. We have also updated the Methods section to explain how we integrated information from the recombinant strains in building the refined set of kinetic models.

Finally, our detailed point-by-point responses to the reviewers' comments are highlighted in blue below.

The revised manuscript is available in two versions: a clean copy and a version with all alterations highlighted in blue.

Reviewer #1 (Remarks to the Author):

I think this is a better version of the manuscript, most of my comments were addressed. However, all the reviewers brought the concern of more use cases to further validate the approach. This doesn't need to be experimental, they can find other published cases as they did with the anthranilate case. The authors argue that some experimental groups have already expressed interest in using their framework. I still think that this work would be much better presented if it presents more use cases.

As suggested by the reviewer, we conducted an additional computational case study. For details, please refer to our response to all reviewers above and the new sections in the main manuscript.

Reviewer #2 (Remarks to the Author):

The authors have addressed my comments largely through text responses as opposed to additional analyses, but their sensitivity analysis on the degree to which the solutions can deviate from the reference point is appreciated. It does raise an additional question though - they find that larger perturbations result in poorer designs if I understand correctly, but are they sure that this isn't simply a result of relying on log-linear kinetics that are increasingly inaccurate for larger perturbations, or dependent on their predictions of growth rate based on the lumped reaction, which could have similar issues for larger perturbations? i.e. Is this not a problem with the limitations of their workflow more so than with large perturbations in general? The details of the designs were not discussed, so it's difficult for me to determine the biochemistry of what is problematic in the large perturbation designs.

In addressing these questions, we want first to stress two general points: (i) deviating significantly from robust physiology while striving for enhanced target chemical production can be detrimental to the organism, as discussed in the manuscript. (ii) regardless of model quality and the use of log-linear kinetics, some designs can be effective for small perturbations while detrimental for larger ones because of the intrinsic nonlinear nature of cellular metabolism.

The reviewer is correct in pointing out that the poor performance of a design under larger perturbations can also be due to the inaccuracies inherent to log-linear models, as we have demonstrated in Figure 3, main text, and Supplementary Note VI. However, it is not evident whether the inferior performance of the designs under larger perturbations is due to inaccurate log-linear predictions, due to (i) and (ii) above, or a combination of these three factors.

One way to identify designs with larger perturbations is to use nonlinear simulations to assess their performance in an exhaustive manner. However, as discussed in the main manuscript, this approach is computationally impractical. The underlying design optimization problem is intricate, encompassing large-scale, highly nonlinear systems and necessitating compliance with dynamic performance

specifications. The most common approach to address this complexity often involves approximating the solution space, as done here with the log-linear approximation, before proceeding with the solution. Therefore, we view our rational strain design approach as a feature, not a flaw.

Conceptually, our workflow balances between accuracy (offered by nonlinear simulations) and computational efficiency (associated with log-linear kinetics). It overcomes the limited accuracy posed by log-linear approximations by using smaller, more constrained steps that provide more robust results in a nonlinear setting. Because we have access to efficient, accurate, and well-established tools for predicting metabolic responses for small perturbations, such as Network Response Analysis (Tsouka et al., 2021), we retain the prediction accuracy while not perturbing the physiology too much. Such a philosophy of cautiously improving the system performance is similar to the windsurfer approach in the systems and control theory (see, e.g., Lee et al., *Automatica*, Vol. 31, No. 11. pp.1619-1636, 1995). In the broader context of design-build-test-learn cycles, it might be advantageous to make incremental but reliable improvements, use experimental data to calibrate models further, and then conduct another computational study to design subsequent strains.

The remainder of their responses are logical but still feel overconfident given the thin validation. The workflow is still interesting and the idea of minimizing metabolic adjustment during designs is logical, but without further validation and given the continued numerical limitations (very few models evaluated in the end to assess alternate designs, dependence on linear control coefficients), it still feels unproven.

As stated above in the response to all reviewers, we have conducted an additional study to demonstrate the workflow potential further. Please consult the revised Results section and in particular, the new subsection of the manuscript that details this study.

Regarding the reviewer's remark on very few models evaluated to assess alternate designs, we would like to stress that this is one of the principal motivations of our study. As we have argued in the main text, conducting a comprehensive, brute-force simulation using large-scale nonlinear models would be prohibitively time-consuming, even with the most performant computers. One approach could involve an ad-hoc selection of a significantly reduced set of candidate target enzymes, followed by simulations on this reduced selection. However, this approach carries the risk of overlooking non-obvious yet critical targets.

Our study addresses this challenge by utilizing log-linear control coefficients within an optimization framework. This method allows us to intelligently select a reduced set of targets, considering the network effects of all enzymes, increasing the likelihood of identifying effective designs.

I think this work will still attract a lot of interest in the community because of the scale of modeling that is now possible - However, on the other hand I hope that kinetic modelers begin to take validation seriously because these models continue to grow in complexity and scale, making evaluation of the multitude of workflow decisions near impossible.

We could not agree more with the reviewers' statements. Since the scale and properties of recent kinetic models are reaching the quality required for implementing the learnings from these models in experimental studies, we and other kinetic modelers are actively taking part and searching for collaborations that would allow us to validate these models. Along these lines, we are collaborating on a separate study aimed at improving the performances of *S. cerevisiae* in producing a high-value compound.

Regarding the reviewer's remark about the multitude of workflow decisions – this is one of the major motivations that instigated this work because we wanted to produce a workflow with sound criteria for choosing the right design decisions.

Reviewer #3 (Remarks to the Author):

I greatly appreciate the effort made by the authors to respond to my previous comments, but I feel that my main concerns related to this article still remain. Despite the complexity and complete computational work of the study, the final results, in terms of quantitative accuracy and validation, are, in my opinion, insufficient to recommend its publication in the current form.

We have provided an additional study to address the reviewer's concerns about quantitative accuracy and validation. For details, please see our response to all reviewers and the substantially revised Results section, including the newly introduced subsections in the main manuscript.

Regarding the lack of quantitative accuracy, I agree with the authors that MONAD captures qualitative trends, but for high-quality kinetic models, it would be expected more accurate quantitative predictions. In this sense, I appreciate the authors' additional analysis presented in Supplementary Note VIII, which identifies the source of discrepancies between model predictions and experimental data. However, rather than alleviating my concerns, this analysis has somewhat heightened them. The need for additional reactions to be overexpressed to fit the experimental data raises questions about whether the models used are overly constrained. Could this reduced degree of freedom in the models hinder the identification of better designs?

As explained above in our general response to all reviewers, we obtained new models with increased prediction accuracy by integrating additional experimental data. We provided details in the new section of the manuscript.

In response to concerns about the models being overly constrained, it is essential to note that the limitation of models built solely with wild-type data in accurately predicting experimental responses of recombinant strains arises from the significant uncertainty within the parameter space. Indeed, multiple parameter sets can describe wild-type responses, but only a subset of these sets is consistent with recombinant strain responses. A comprehensive exploration of the parameter space is required to identify this subset. Alternatively, as demonstrated in our new study, integrating additional data can help identify such parameter sets more efficiently. The existence of numerous parameter sets that can accurately describe both wild-type and recombinant experimental responses suggests that the models are not overly constrained.

On the other hand, one might argue that constraints on fluxes, concentrations, and enzyme levels could hinder the identification of more optimal designs. However, these constraints yield designs that may be less optimal but are more likely to be successful in experimental implementation, as we have thoroughly explained in the manuscript.

In terms of validation, I still believe that a proper experimental validation of NOMAD designs would be desirable (only 3 genetic interventions are needed). While I understand that experimental validation may not be the primary focus of this work, I agree with reviewer 1 that addressing, at least one more case study, would enhance the scope and value of the proposed workflow.

Agree with the authors that rational strain design using kinetic models is really a holy grail in metabolic engineering. Therefore, I commend the authors for the tremendous effort done in this sense. However, I firmly believe that a more robust validation, either experimental or computational, is necessary to strengthen the findings.

We concur with the reviewer that experimental validations of computational workflows such as NOMAD are necessary for advancing the field and facilitating the routine employment of such techniques in bioengineering applications. However, the computational groups can only perform even the simplest experimental validations in collaboration with the experimental groups, which frequently can prove difficult, as witnessed by a relatively small number of publications with such

collaborations. As mentioned in our response to reviewer 2, we are currently working on a project of such nature; however, that work falls outside the scope of this manuscript. In response to the reviewer's suggestion, we have included an additional computational case study instead.

References

- (1) Lee, K. H.; Park, J. H.; Kim, T. Y.; Kim, H. U.; Lee, S. Y. Systems Metabolic Engineering of Escherichia Coli for L-Threonine Production. *Mol. Syst. Biol.* **2007**, *3* (1), 149. <https://doi.org/10.1038/msb4100196>.
- (2) Ahn, J.; Chung, B. K. S.; Lee, D.-Y.; Park, M.; Karimi, I. A.; Jung, J.-K.; Lee, H. NADPH-Dependent Pgi-Gene Knockout Escherichia Coli Metabolism Producing Shikimate on Different Carbon Sources. *FEMS Microbiol. Lett.* **2011**, *324* (1), 10–16. <https://doi.org/10.1111/j.1574-6968.2011.02378.x>.
- (3) Gu, P.; Yang, F.; Kang, J.; Wang, Q.; Qi, Q. One-Step of Tryptophan Attenuator Inactivation and Promoter Swapping to Improve the Production of L-Tryptophan in Escherichia Coli. *Microb. Cell Factories* **2012**, *11* (1), 30. <https://doi.org/10.1186/1475-2859-11-30>.
- (4) Jun Choi, Y.; Hwan Park, J.; Yong Kim, T.; Yup Lee, S. Metabolic Engineering of Escherichia Coli for the Production of 1-Propanol. *Metab. Eng.* **2012**, *14* (5), 477–486. <https://doi.org/10.1016/j.ymben.2012.07.006>.
- (5) Liu, Q.; Cheng, Y.; Xie, X.; Xu, Q.; Chen, N. Modification of Tryptophan Transport System and Its Impact on Production of L-Tryptophan in Escherichia Coli. *Bioresour. Technol.* **2012**, *114*, 549–554. <https://doi.org/10.1016/j.biortech.2012.02.088>.
- (6) Gu, P.; Yang, F.; Li, F.; Liang, Q.; Qi, Q. Knocking out Analysis of Tryptophan Permeases in Escherichia Coli for Improving L-Tryptophan Production. *Appl. Microbiol. Biotechnol.* **2013**, *97* (15), 6677–6683. <https://doi.org/10.1007/s00253-013-4988-5>.
- (7) Luo, W.; Huang, J.; Zhu, X.; Huang, L.; Cai, J.; Xu, Z. Enhanced Production of L-Tryptophan with Glucose Feeding and Surfactant Addition and Related Metabolic Flux Redistribution in the Recombinant Escherichia Coli. *Food Sci. Biotechnol.* **2013**, *22* (1), 207–214. <https://doi.org/10.1007/s10068-013-0029-5>.
- (8) Lin, Y.; Sun, X.; Yuan, Q.; Yan, Y. Extending Shikimate Pathway for the Production of Muconic Acid and Its Precursor Salicylic Acid in Escherichia Coli. *Metab. Eng.* **2014**, *23*, 62–69. <https://doi.org/10.1016/j.ymben.2014.02.009>.
- (9) Liu, S. P.; Liu, R. X.; Xiao, M. R.; Zhang, L.; Ding, Z. Y.; Gu, Z. H.; Shi, G. Y. A Systems Level Engineered E. Coli Capable of Efficiently Producing L-Phenylalanine. *Process Biochem.* **2014**, *49* (5), 751–757. <https://doi.org/10.1016/j.procbio.2014.01.001>.
- (10) Kim, B.; Park, H.; Na, D.; Lee, S. Y. Metabolic Engineering of Escherichia Coli for the Production of Phenol from Glucose. *Biotechnol. J.* **2014**, *9* (5), 621–629. <https://doi.org/10.1002/biot.201300263>.
- (11) Gu, P.; Yang, F.; Su, T.; Li, F.; Li, Y.; Qi, Q. Construction of an L-Serine Producing Escherichia Coli via Metabolic Engineering. *J. Ind. Microbiol. Biotechnol.* **2014**, *41* (9), 1443–1450. <https://doi.org/10.1007/s10295-014-1476-6>.
- (12) Cui, Y.-Y.; Ling, C.; Zhang, Y.-Y.; Huang, J.; Liu, J.-Z. Production of Shikimic Acid from Escherichia Coli through Chemically Inducible Chromosomal Evolution and Cofactor Metabolic Engineering. *Microb. Cell Factories* **2014**, *13* (1), 21. <https://doi.org/10.1186/1475-2859-13-21>.
- (13) Cheol Kim, S.; Eun Min, B.; Gyu Hwang, H.; Woo Seo, S.; Yeol Jung, G. Pathway Optimization by Re-Design of Untranslated Regions for L-Tyrosine Production in Escherichia Coli. *Sci. Rep.* **2015**, *5* (1), 13853. <https://doi.org/10.1038/srep13853>.
- (14) Ding, D.; Liu, Y.; Xu, Y.; Zheng, P.; Li, H.; Zhang, D.; Sun, J. Improving the Production of L-Phenylalanine by Identifying Key Enzymes Through Multi-Enzyme Reaction System in Vitro. *Sci. Rep.* **2016**, *6* (1), 32208. <https://doi.org/10.1038/srep32208>.

- (15) Liu, X.; Lin, J.; Hu, H.; Zhou, B.; Zhu, B. Site-Specific Integration and Constitutive Expression of Key Genes into Escherichia Coli Chromosome Increases Shikimic Acid Yields. *Enzyme Microb. Technol.* **2016**, *82*, 96–104. <https://doi.org/10.1016/j.enzmictec.2015.08.018>.
- (16) Liu, L.; Chen, S.; Wu, J. Phosphoenolpyruvate:Glucose Phosphotransferase System Modification Increases the Conversion Rate during L-Tryptophan Production in Escherichia Coli. *J. Ind. Microbiol. Biotechnol.* **2017**, *44* (10), 1385–1395. <https://doi.org/10.1007/s10295-017-1959-3>.
- (17) Gupta, A.; Reizman, I. M. B.; Reisch, C. R.; Prather, K. L. J. Dynamic Regulation of Metabolic Flux in Engineered Bacteria Using a Pathway-Independent Quorum-Sensing Circuit. *Nat. Biotechnol.* **2017**, *35* (3), 273–279. <https://doi.org/10.1038/nbt.3796>.
- (18) Xu, Q.; Bai, F.; Chen, N.; Bai, G. Gene Modification of the Acetate Biosynthesis Pathway in Escherichia Coli and Implementation of the Cell Recycling Technology to Increase L-Tryptophan Production. *PLOS ONE* **2017**, *12* (6), e0179240. <https://doi.org/10.1371/journal.pone.0179240>.
- (19) Chen, L.; Chen, M.; Ma, C.; Zeng, A.-P. Discovery of Feed-Forward Regulation in L-Tryptophan Biosynthesis and Its Use in Metabolic Engineering of E. Coli for Efficient Tryptophan Bioproduction. *Metab. Eng.* **2018**, *47*, 434–444. <https://doi.org/10.1016/j.ymben.2018.05.001>.
- (20) Liu, Y.; Xu, Y.; Ding, D.; Wen, J.; Zhu, B.; Zhang, D. Genetic Engineering of Escherichia Coli to Improve L-Phenylalanine Production. *BMC Biotechnol.* **2018**, *18* (1), 5. <https://doi.org/10.1186/s12896-018-0418-1>.
- (21) Martínez, J. A.; Rodríguez, A.; Moreno, F.; Flores, N.; Lara, A. R.; Ramírez, O. T.; Gosset, G.; Bolívar, F. Metabolic Modeling and Response Surface Analysis of an Escherichia Coli Strain Engineered for Shikimic Acid Production. *BMC Syst. Biol.* **2018**, *12* (1), 102. <https://doi.org/10.1186/s12918-018-0632-4>.
- (22) Li, Z.; Wang, H.; Ding, D.; Liu, Y.; Fang, H.; Chang, Z.; Chen, T.; Zhang, D. Metabolic Engineering of Escherichia Coli for Production of Chemicals Derived from the Shikimate Pathway. *J. Ind. Microbiol. Biotechnol.* **2020**, *47* (6–7), 525–535. <https://doi.org/10.1007/s10295-020-02288-2>.

Reviewers' Comments:

Reviewer #1:

Remarks to the Author:

My main comment of the need of another case of validation has been addressed. This is a much better version of the work. I can endorse the publication of this manuscript

Reviewer #2:

Remarks to the Author:

I have read the revised manuscript, most notably the new validation exercise presented in Figure 6. I think the authors have done an admirable job to try to validate their model given limitations in generating new experimental data. I have no further concerns.

Reviewer #3:

Remarks to the Author:

Narayanan and colleagues present the third iteration of their manuscript titled "Rational Strain Design with Minimal Phenotype Perturbation." The second submission effectively tackled major conceptual and methodological weaknesses, yet my concerns regarding the limited validation of their workflow remain partially addressed in this latest revision.

The authors introduce an improved workflow, leveraging additional data to achieve significantly more accurate quantitative predictions. However, I remain cautious about considering this as a distinct validation process, as it builds upon the same case study. The authors argue that the scarcity of available data hinders the exploration of an entirely new case study. Additionally, the experimental validation is not considered due to the computational nature of the team and because they plan for such validation outlined in a subsequent collaboration with an experimental group.

Overall, and considering the piece's evident interest to the broader community and its technical and conceptual coherence, I find no further objections to its publication.